# Dishevelled-3 conformation dynamics analyzed by FRET-based biosensors reveals a key role of casein kinase 1

Jakub Harnoš[1,11], Maria Consuelo Alonso Cañizal [2,3,4], Miroslav Jurásek[5,6], Jitender Kumar [5], Cornelia Holler [7,8], Alexandra Schambony [7,8], Kateřina Hanáková[5,6], Ondřej Bernatík[1], Zbyněk Zdráhal[5,6], Kristína Gömöryová [1], Tomáš Gybel[1], Tomasz Witold Radaszkiewicz[1], Marek Kravec[1], Lukáš Trantírek [5,9], Jan Ryneš[5], Zankruti Dave [1], Ana Iris Fernández-Llamazares [10], Robert Vácha[5,6], Konstantinos Tripsianes [5], Carsten Hoffmann [2,3,4] & Vítězslav Bryja [1,9]

Dishevelled (DVL) is the key component of the Wnt signaling pathway. Currently, DVL conformational dynamics under native conditions is unknown. To overcome this limitation, we develop the Fluorescein Arsenical Hairpin Binder- (FlAsH-) based FRET in vivo approach to study DVL conformation in living cells. Using this single-cell FRET approach, we demonstrate that (i) Wnt ligands induce open DVL conformation, (ii) DVL variants that are predominantly open, show more even subcellular localization and more efficient membrane recruitment by Frizzled (FZD) and (iii) Casein kinase 1 ε (CK1ε) has a key regulatory function in DVL conformational dynamics. In silico modeling and in vitro biophysical methods explain how CK1ε-specific phosphorylation events control DVL conformations via modulation of the PDZ domain and its interaction with DVL C-terminus. In summary, our study describes an experimental tool for DVL conformational sampling in living cells and elucidates the essential regulatory role of CK1ε in DVL conformational dynamics.

[1] Department of Experimental Biology, Faculty of Science, Masaryk University, Brno 62500, Czech Republic. [2] Department of Pharmacology and Toxicology, University of Würzburg, Würzburg 97078, Germany. [3] Rudolf Virchow Center for Experimental Biomedicine, University of Würzburg, Würzburg 97078, Germany. [4] Institute for Molecular Cell Biology, CMB—Center for Molecular Biomedicine, University Hospital Jena, Friedrich Schiller University Jena, Jena 07745, Germany. [5] CEITEC—Central European Institute of Technology, Masaryk University, Brno 62500, Czech Republic. [6] National Centre for Biomolecular Research, Faculty of Science, Masaryk University, Brno, 62500, Czech Republic. [7] Max Planck Institute for the Science of Light, Erlangen 91058, Germany. [8] Biology Department, Developmental Biology, Friedrich-Alexander University Erlangen-Nüremberg, Erlangen 91058, Germany. [9] Institute of Biophysics, Academy of Sciences of the Czech Republic, v.v.i., Brno 612 65, Czech Republic. [10] Pepscan Therapeutics B.V., Lelystad 8243, The Netherlands. [11] Present address: Department of Cell, Developmental & Regenerative Biology, Icahn School of Medicine at Mount Sinai, New York, NY 10029, USA. Correspondence and requests for materials should be addressed to V.B. (email: bryja@sci.muni.cz)

Wnt signaling pathway is one of the signaling pathways, whose dysfunction or deregulation is linked to developmental defects, inherited diseases, and many types of cancer[1]. Wnt proteins have been shown to activate several Wnt signaling pathways. The best known Wnt/β-catenin (canonical) pathway typically controls cell proliferation and differentiation by the regulation of transcription, whereas other (i.e. non-canonical) branches typically regulate cell polarity and migration by the reorganization of cytoskeleton[2–4]. Wnt signals are transduced via the membrane receptor Frizzled (FZD) and the intracellular protein Dishevelled (Dsh in *Drosophila*, DVL1–3 in humans). There is a general agreement based on strong genetic evidence in *Drosophila*[3] and mouse[5,6] that DVL protein(s) plays a crucial role in all major branches of Wnt signaling and serves as a signaling hub.

All DVL proteins have a conserved architecture consisting of three well-defined domains: an N-terminal DIX domain (Disheveled and Axin), a central PDZ domain (Post-synaptic Density Protein-95, Disc Large Tumor Suppressor, and Zonula Occludens-1), and a C-terminal DEP domain (Disheveled, Egl-10, and Pleckstrin). Crystal structures of all three isolated DVL domains are known[7–9]. The regions linking the three domains and the long C-terminus (200 aa) are intrinsically disordered[2].

DVL molecules can multimerize to form high-order structures called DVL signalosomes, which are crucial for the subsequent downstream activation of Wnt/β-catenin signaling[10–12]. Signalosome formation is mediated via head-to-tail multimerization of DVL's DIX domains[10] (and/or DIX domain of Axin protein[11–13]). Recently, dimerization via the DEP domain has also been reported and it was proposed that DEP–DEP interaction mediated by domain swapping cooperates with DIX-dependent DVL signalosome formation[14].

Several pieces of information suggest that DVL can also interact intramolecularly. Specifically, it has been shown in vitro that at least two regions located in the C-terminus of DVL— (i) a nuclear export signal and (ii) the terminal 7 aa in the C-terminus of DVL—can interact with the PDZ domain[15–17]. These reports suggested that DVL can exist in a closed and open conformation(s) undergoing multiple structural transitions that control DVL function. However, there is neither an experimental approach to study this phenomenon directly nor clear evidence that DVL exists in multiple conformations in vivo.

In this study we use an experimental system that enables the analysis of DVL conformational sampling in living mammalian cells. We design and characterize several biosensors of DVL3 conformation based on fluorescein arsenical hairpin binder (FlAsH)-based Förster resonance energy transfer (FRET). This single-cell FRET method allows site-selective detection of protein conformation dynamics in living cells and has already helped to decipher various biological problems[18–20]. FlAsH is a small organic compound, which recognizes and binds to CCPGCC tag[18]. The FlAsH–CCPGCC complex becomes fluorescent and can be used as an energy acceptor in the FRET pair with ECFP (enhanced cyan fluorescent protein). The main advantage of the CCPGCC tag is its size—only 6 aa, which reduces interference with the biological function of the protein of interest[18]. Using FlAsH FRET DVL3 sensors we describe the conformational dynamics of DVL3 in living cells upon Wnt pathway stimulation and the key function of Casein kinase 1 ε (CK1ε) in this process.

## Results

### Design and generation of the FRET-based DVL3 biosensors.
FlAsH-based FRET represents a powerful technique that allows monitoring of the protein conformation in biological setups, where large tags can interfere with protein complex formation and function[18–20]. FlAsH molecule forms a fluorescent complex with a 6 aa long sequence (CCPGCC) that can be shut off by addition of BAL (British Anti-Lewisite or 2,3-dimercapto-1-propanol)[18]. The general scheme of how FlAsH-based FRET works is depicted in Fig. 1a and in Supplementary Fig. 1a. We designed and generated four different DVL3 sensors (FlAsH I–IV), which differ in the position of the CCPGCC tag. Each DVL3 sensor contained the ECFP tag at the N-terminus and one CCPGCC internal tag, located in an intrinsically disordered region (IDR) or at the C-terminus of DVL3. The CCPGCC tags were inserted at the most disordered region of each IDR predicted in silico using PONDR-FIT tool[21] as shown in Fig. 1b. Biological and signaling properties of all four DVL3 FlAsH sensors were indistinguishable from wild-type (wt) ECFP-DVL3 in the following assays (performed in HEK293 DVL1-2-3$^{-/-}$ cell line[22] in order to avoid interaction with endogenous full-length DVL isoforms): activation of the Wnt/β-catenin pathway monitored by Dual Luciferase TopFlash/Renilla reporter gene assay (Supplementary Fig. 1c); CK1ε-dependent DVL electrophoretic mobility shift assay (Supplementary Fig. 1d); and changes of DVL subcellular localization induced by CK1ε (Supplementary Fig. 1e). Data are summarized in Fig. 1c.

The ECFP-DVL3 FlAsH I, II, and III (but not IV) sensors showed robust basal intramolecular FRET measure after BAL addition (Fig. 1d, left panel), which suggests that the ECFP and CCPGCC tags are in physical proximity. Since it has been reported that DVL can undergo intermolecular multimerization via its DIX and DEP domains[10,14], intermolecular FRET among different DVL3 molecules can also occur. To assess the intermolecular FRET efficiency, we generated N-terminally HA-tagged DVL3 FlAsH I–IV constructs without the N-terminal ECFP tag. Despite the fact that the ECFP and CCPGCC tags (labeled by FlAsH molecule) showed almost complete co-localization in dots for all four sensors (Supplementary Fig. 1b), the intermolecular FRET efficiency was negligible, except FlAsH III (Fig. 1d, right panel). These results demonstrate that ECFP-DVL3 FlAsH I, II, and III sensors represent a useful tool for the analysis of DVL conformation in living cells that is not affected (FlAsH I, II) or only minimally affected (FlAsH III) by intermolecular FRET due to the DVL–DVL oligomerization.

### CK1ε changes intramolecular FRET in DVL3 sensors.
As a result of Wnt pathway activation, DVL is heavily phosphorylated[23–26]. It has been further shown that the major DVL kinase responsible for Wnt-induced phosphorylation of DVL[27–29] is CK1ε. In order to analyze whether and how CK1ε affects DVL3 conformation, we (i) used CK1δ/ε inhibitor PF670462 (refs. [29,30]) to pharmacologically inhibit endogenous CK1ε and (ii) promoted DVL3 phosphorylation by overexpression of exogenous CK1ε. Such manipulation of CK1ε activity dramatically affects DVL3 properties such as: (i) CK1ε-DVL3-mediated activation of the Wnt/β-catenin pathway monitored by Dual Luciferase TopFlash/Renilla reporter gene assay (Fig. 2a); (ii) phosphorylation status of DVL3 assessed either as a phosphorylation-dependent mobility shift on SDS-PAGE or as an increased signal for pS643-DVL3 phosphorylation-specific antibody—a well-defined[29] target site for CK1ε (Fig. 2b), and (iii) changes in the subcellular localization of DVL3 (ref. [31]) analyzed by immunofluorescence (Fig. 2c). These properties were analyzed for all four ECFP-DVL3 FlAsH sensors (Supplementary Fig. 2a) and were found indistinguishable from wt FLAG-DVL3 (compare Supplementary Fig. 2a and Fig. 2a–c). This result shows that neither ECFP nor CCPGCC tag addition in any of the four sensors affects the biological response of DVL3 to CK1ε.

To determine how CK1ε influences DVL3 conformation sampling, we analyzed basal FRET in ECFP-DVL3 FlAsH I–IV

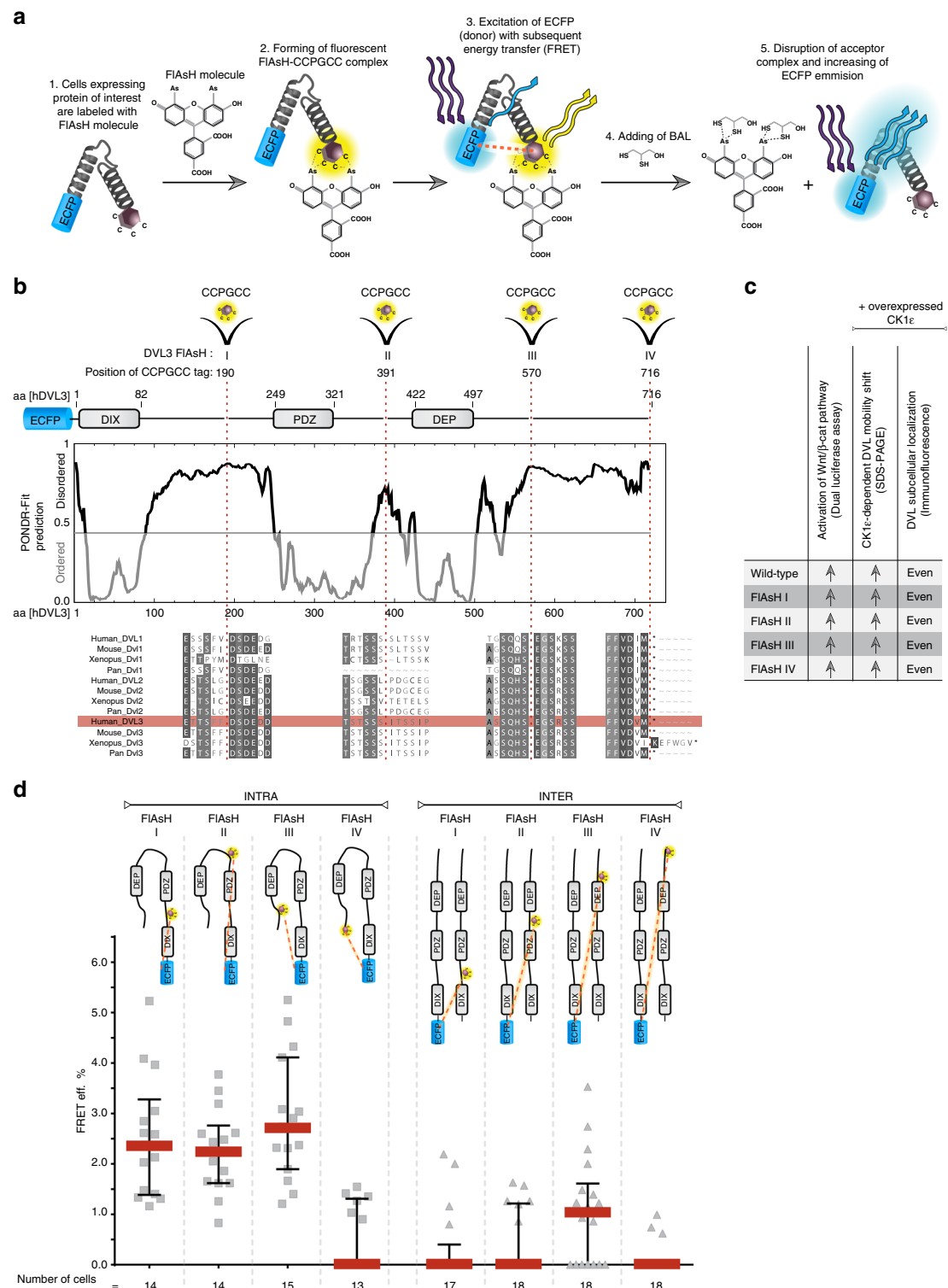

**Fig. 1** The FlAsH-based FRET DVL3 sensors. **a** The general scheme of the FlAsH-based FRET in vivo approach (in further detail in Supplementary Fig. 1a). **b** The design of four DVL3 FlAsH I–IV sensors with the CCPGCC tag and N-terminal ECFP tag. The insertions of the CCPGCC tag were placed at the positions with highest disordered prediction scores (PONDR-Fit[21])—one CCPGCC tag per a linker region (FlAsH I, II, III) and the C-terminus (FlAsH IV). Multiple sequence alignment of the Dvl/DVL sequences at the site of insertion is shown below. Residues with >80% similarity are highlighted. Sequence of human DVL3 used for cloning as a template is shown in red box. **c** Biological properties of four ECFP-DVL3 FlAsH sensors are indistinguishable from wild-type ECFP-DVL3 (for details see Supplementary Fig. 1c–e). **d** The intramolecular and intermolecular FRET efficiency in the DVL3 FlAsH I–IV sensors in HEK293 wild-type cells. The position of the fluorophores in DVL3 molecules in both experimental setups are schematized above the graph. One data point corresponds to one analyzed cell. Data from three independent transfections were merged. Data in **d** represent median ± interquartile range. FRET eff., FRET efficiency; BAL, British anti-Lewisite; ECFP enhanced cyan fluorescent protein

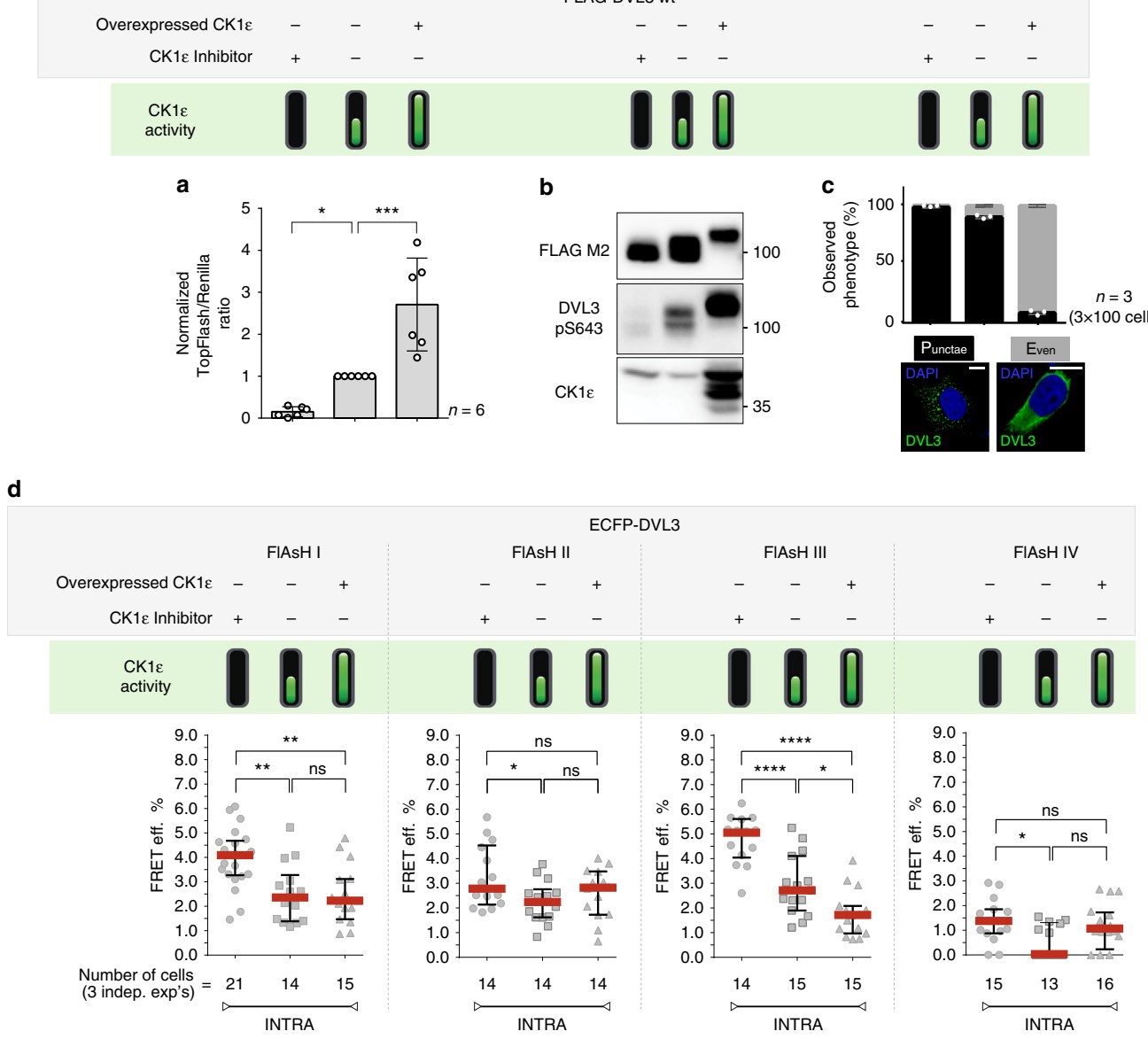

**Fig. 2** DVL3 FlAsH I–IV sensors uncover the role of CK1ε in DVL3 conformation. **a** The effects of the CK1δ/ε inhibitor PF-670642 and the overexpressed CK1ε on the downstream Wnt/β-catenin signaling of FLAG-DVL3 was quantified by TopFlash assay, **b** phosphorylation status detected either as phosphorylation-dependent mobility shift on SDS-PAGE or as an increased signal for pS643-DVL3 phosphorylation-specific antibody, and **c** changes in the localization of DVL3 (punctae phenotype plotted as white dots); DAPI was used for nuclei staining; scale bars: 10 μm. **d** Measurements of the intramolecular FRET efficiency of the ECFP-DVL3 FlAsH sensors I–IV in HEK293 wild-type cells are shown. One data point corresponds to one analyzed cell; datapoints from three independent transfections were merged. Data in **a** and **c** represent mean ± S.D., data in **d** represent median ± interquartile range. Statistical significance in **a/d** was analyzed by one-way ANOVA test with Gaussian distribution and Tukey's post-test (*$p \leq 0.05$; **$p \leq 0.01$; ***$p \leq 0.001$; ****$p \leq 0.0001$; ns, not significant, $p > 0.05$)

sensors in: (i) cells treated with the CK1δ/ε inhibitor (PF670462, 10 μM), (ii) control cells possessing active endogenous CK1ε, and (iii) cells overexpressing CK1ε (Fig. 2d, for intermolecular FRET efficiency controls see Supplementary Fig. 2b). Changes in CK1ε levels and activity affected FRET efficiency to a different extent for each sensor. The most prominent differences were detected using the FlAsH III sensor, for which the CK1δ/ε inhibitor consistently increased the FRET efficiency, whereas the CK1ε overexpression decreased it. These experiments implied that CK1ε inhibition results in a more compact (i.e. closed) conformation of DVL3, while CK1ε overexpression leads to a looser (i.e. open) conformation. In order to dissect

mechanistically how CK1ε affects the DVL conformations in vivo, we selected DVL3 FlAsH III sensor for a rigorous investigation.

**Identification of DVL3 regions in direct contact with CK1ε.** As a next step, we mapped in detail the interaction site(s) of DVL3 and CK1ε, in order to generate DVL3 variants incapable of CK1ε binding while keeping all structured domains intact. We hypothesized that IDRs significantly contribute to the interaction of DVL3 and CK1ε. The binding epitopes within IDRs are typically defined by linear peptide motifs[32] that can be rapidly analyzed in

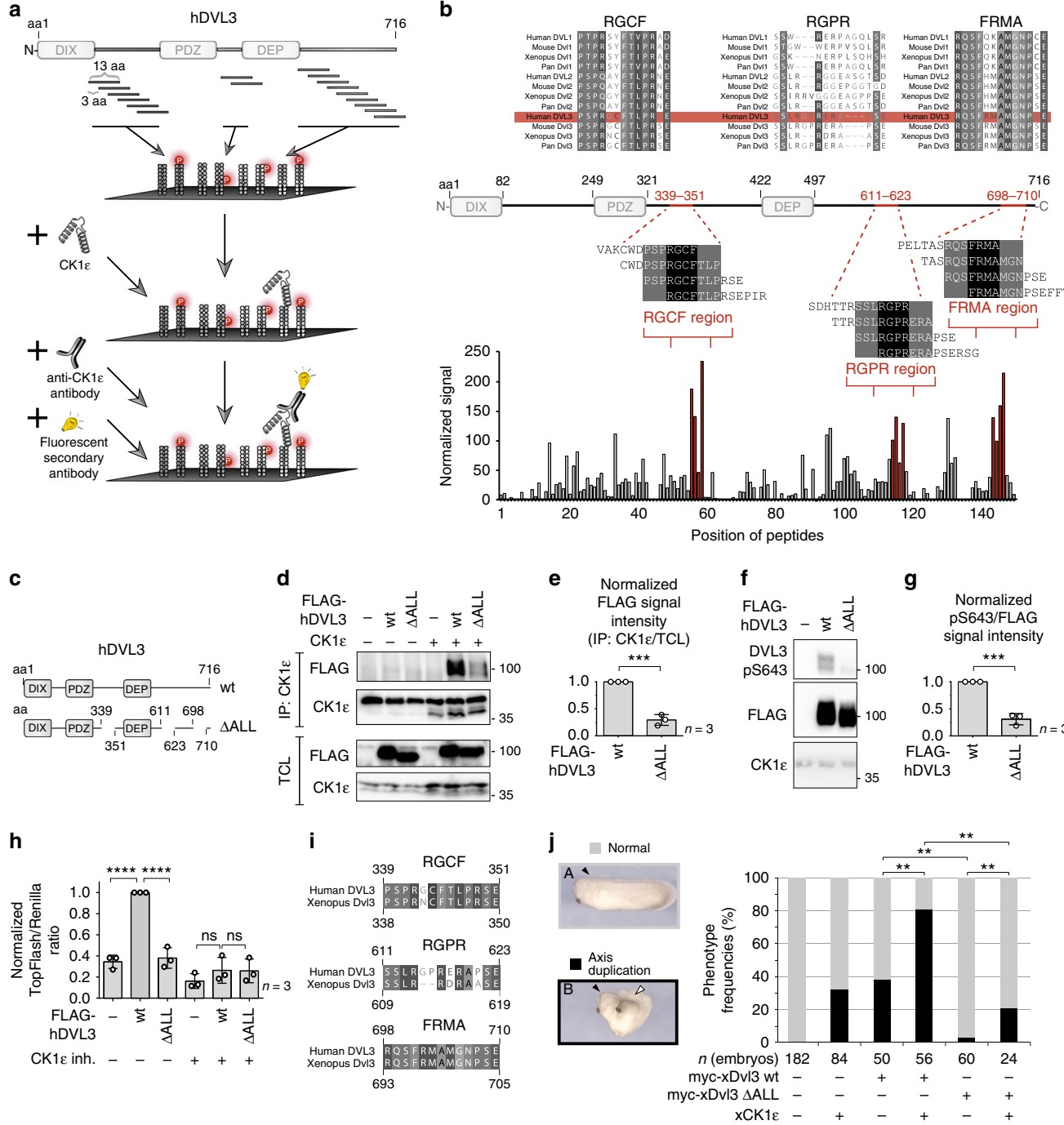

**Fig. 3** The molecular and functional analysis of DVL3 regions interacting with CK1ε. **a** The general workflow of a peptide array analysis: immobilized peptides were incubated with CK1ε, then with anti-CK1ε antibody followed by fluorescent secondary antibodies, and detected by reading the fluorescence intensity. Peptide array contained the non-modified or phosphorylated peptide variants from the intrinsically disordered regions (IDRs) according to phosphorylation pattern by CK1ε, as mapped earlier[29] (see Supplementary Fig. 3). **b** Identification of three regions (named RGCF, RGPR, and FRMA regions by their central 4 aa sequences), which bind CK1ε with high affinity. Multiple sequence alignment for the RGCF, RGPR, and FRMA regions of various Dvl/ DVL isoforms is shown above. Residues with >80% similarity are highlighted and human DVL3 sequence is denoted by red box; only non-modified (i.e. non-phosphorylated) peptides are shown in this graph. **c** Generation of the N-terminal FLAG-tagged DVL3 ΔALL variant lacking the interaction interfaces (RGCF, RGPR, FRMA regions) and its subsequent analysis by **d** coimmunoprecipitation and **e** its quantification, **f** by western blot detection of the pS643 phosphorylation level and **f** its quantification, **g** and by Topflash Reporter Assay for the downstream Wnt/β-catenin signaling. **h** The multiple sequence alignment of Xenopus Dvl3 and human DVL3 sequences in the RGCF, RGPR, and FRMA regions is shown. **i** Analysis of the activity of the ΔALL variant derived from Xenopus xDvl3 in the Wnt/β-catenin canonical signaling (in the Xenopus laevis embryos). **j** Left: Representative image of control (low or no activity of the Wnt/β-catenin pathway; in a gray box) or duplicated (high activity; in a black box) axis in the Xenopus laevis embryos. Right: Quantification of the Xenopus laevis embryos with wild-type xDvl3 and the ΔALL variant. Experiments in **d**–**f** were performed in HEK DVL1-2-3$^{-/-}$ cell line. Data in **e**, **g**, **h**, **j** represent mean ± S.D. Data in **h** and **j** were analyzed by one-way ANOVA test with Gaussian distribution; Tukey's post-test was used for statistical analysis (*, $p \leq 0.05$; **, $p \leq 0.01$; ***, $p \leq 0.001$, ****, $p \leq 0.0001$; ns, not significant, $p > 0.05$); data in **e** and **g** were analyzed by Student's t-test

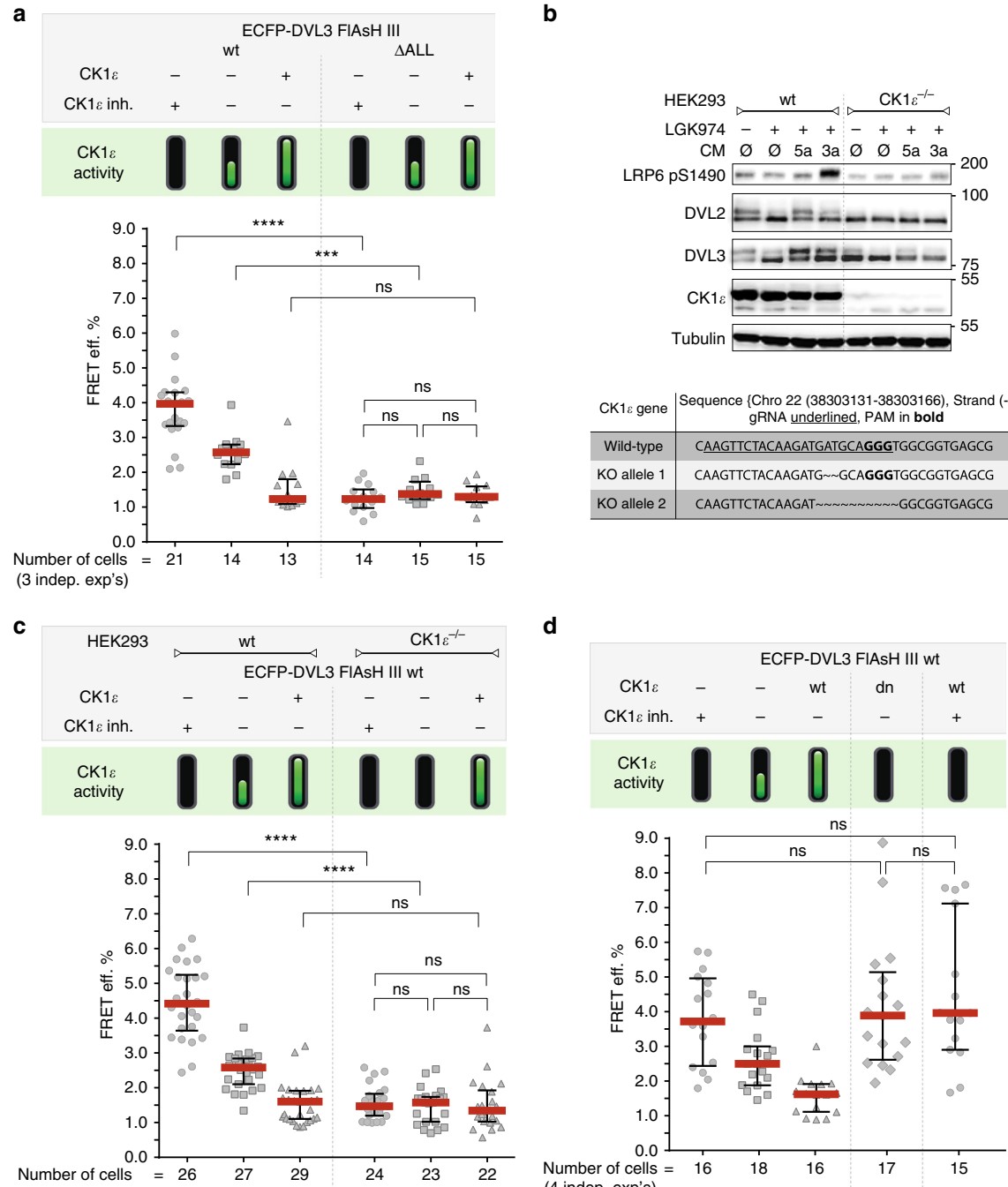

**Fig. 4** CK1ε is required for the conformational dynamics of DVL3. **a** Sequences corresponding to the ΔALL variant were deleted in the ECFP-DVL3 FlAsH III sensor and the intramolecular FRET efficiency measurement of this variant (ECFP-DVL3 FlAsH III ΔALL sensor) in HEK293 wild-type cells is shown. **b** Generation of the CK1ε$^{-/-}$-deficient HEK293 cells using the CRISPR-Cas9 system and analysis of their capacity to respond to the Wnt-3a and Wnt-5a ligands was analyzed by western blotting. Bottom: Sequencing results for CK1ε locus targeted in CK1ε$^{-/-}$ cells are shown; sequences of gRNA, which were used, are underlined, the PAM sequence is in bold. **c** Measurements of the intramolecular FRET efficiency of the wt DVL3 FlAsH sensor III in HEK293 wild type and CK1ε$^{-/-}$-deficient cells. **d** Measurements of the intramolecular FRET efficiency of the wt DVL3 FlAsH III sensor in HEK293 wild-type cells with dominant negative (dn) variant of CK1ε and wt CK1ε treated with the CK1δ/ε inhibitor are shown. Data in **a**, **c** and **d**: one dot represents one cell; data from three to five independent transfections were merged. Median ± interquartile range is indicated; one-way ANOVA test with Gaussian distribution and Tukey's post-test was used for statistical analysis (*, $p \leq 0.05$; **, $p \leq 0.01$; ***, $p \leq 0.001$; ****, $p \leq 0.0001$; ns, not significant, $p > 0.05$). CM: conditional medium, Ø: control, 5a: Wnt-5a, 3a: Wnt-3a

peptide microarrays. For our purposes, we designed a peptide microarray (13-meric peptides overlapping by 10 residues) covering the IDRs of DVL3 (Fig. 3a).

The human DVL3 peptide library also contained phosphorylated versions of those peptides, where phosphorylation sites for

CK1ε were previously identified[29] (Supplementary Fig. 3). After incubation with recombinant CK1ε, we discovered three peptide clusters—named RGCF (aa 339–351), RGPR (aa 611–623), and FRMA (aa 698–710)—as potential interaction sites with CK1ε (Fig. 3b). The MS/MS-based analysis of the wild-type DVL3

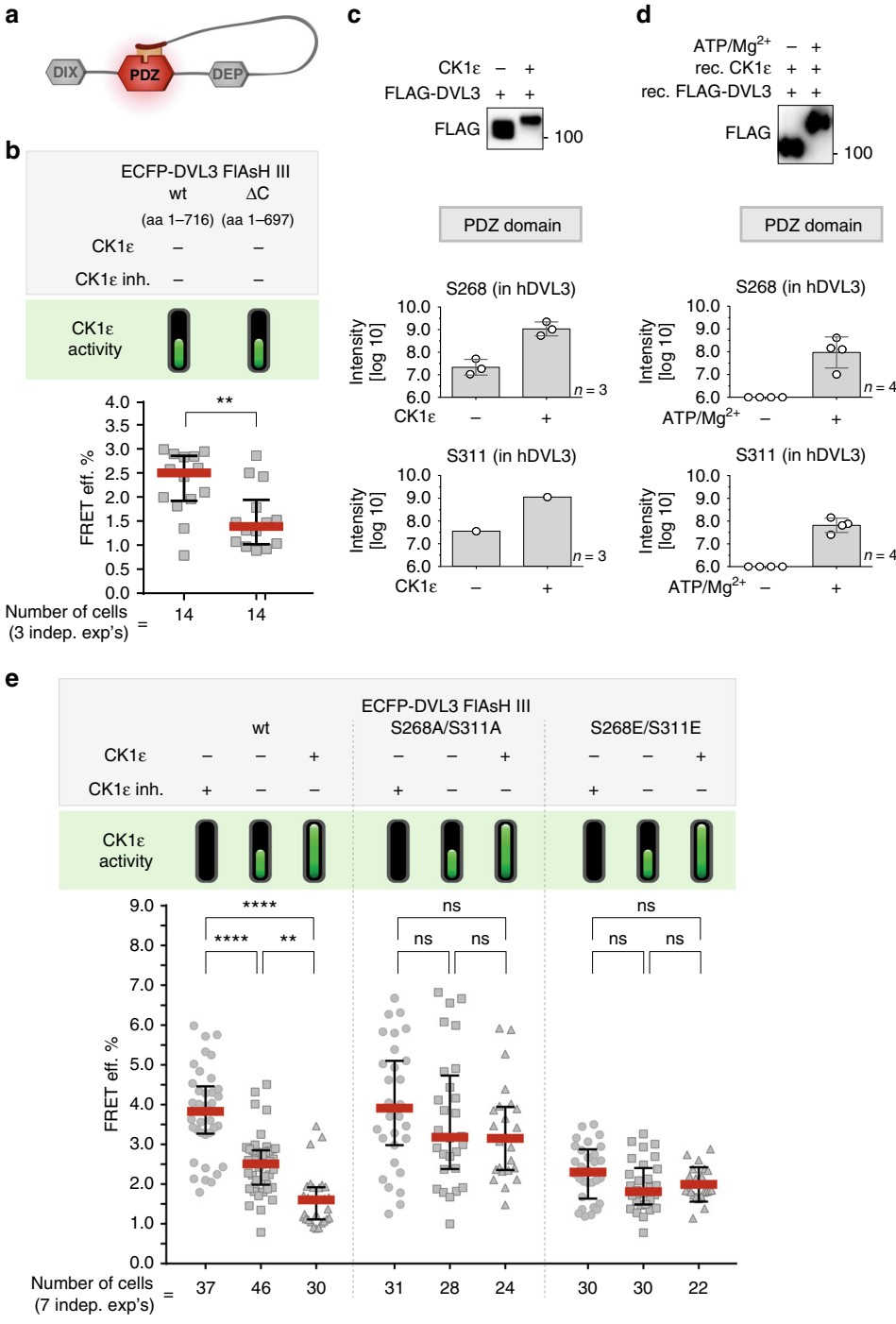

**Fig. 5** Phosphorylation of PDZ domain controls the conformational dynamics of DVL3. **a** Schematic depiction of the closed conformation of DVL proposed here[16], where seven last C-terminal aa (sequence: EFFVDIM) interact with the peptide-binding pocket of the PDZ domain. **b** Measurements of the intramolecular FRET efficiency of the wt DVL3 FlAsH sensors III (aa 1–716 in human DVL3) and the ΔC variant (aa 1–697) in HEK293 wild-type cells. **c** The western blot-based (above) and MS/MS-based (below) analyses of the CK1ε-induced phosphorylation of serine residues present in the PDZ domain of human DVL3. FLAG-DVL3 wt was overexpressed with/without CK1ε wt in HEK293 wt cells, immunoprecipitated, and the level of phosphorylation was analyzed by MS/MS. CK1ε-induced phosphorylation of S268 and S311 in DVL3 PDZ domain was detected. **d** In vitro kinase assay with recombinant FLAG-DVL3 and CK1ε analyzed by western blot (above) and MS/MS (below) confirms that S268 and S311 are direct phosphorylation sites of CK1ε. Values in **c** and **d** show absolute intensity of phosphorylated peptides plotted on a log10 scale. The detection limit is approximately $1.10^6$, i.e. 6.0. Individual datapoints represent biological replicates. **e** Comparison of the intramolecular FRET efficiency of wt DVL3 FlAsH III sensor, non-phosphorylatable variant (S268A/S311A), and phosphorylation-mimicking variant (S268E/S311E) in wt HEK293 cells is shown. One data point corresponds to one analyzed cell; datapoints from 3 (**b**) and 7 (**e**) independent transfections were merged. Data in **b** and **e** represent median ± interquartile range and data in **e** were analyzed by one-way ANOVA test with Gaussian distribution and Tukey's post-test (*, $p \leq 0.05$; **, $p \leq 0.01$; ***, $p \leq 0.001$; ****, $p \leq 0.0001$; ns, not significant, $p > 0.05$). Data in **b** were analyzed by Student's $t$-test. Data in **c** and **d** represent mean ± S.D.

overexpressed in HEK293 cells showed that serine residues in these regions (S350 in RGCF; S611, S612, and S622 in RGPR) were only negligibly phosphorylated, in contrast to S700 in FRMA that was found constitutively phosphorylated (Supplementary Fig. 4b). When CK1ε was overexpressed, all residues in RGCF and RGPR regions were phosphorylated (Supplementary Fig. 4c, d). Importantly, phosphorylated variants of these peptides showed negligible interaction with CK1ε in the peptide array (Supplementary Fig. 4e). The quantification of the in vitro binding between the peptides corresponding to these regions and recombinant CK1ε using fluorescence anisotropy (FA) showed affinities close to 1 μM for all three peptides but negligible interaction for their phosphorylated variants (Supplementary Fig. 4a). This suggests that the interaction is phosphorylation-sensitive and CK1ε is released once the binding epitopes get phosphorylated.

Deletion of these regions in DVL3 (hereafter referred as DVL3 ΔALL variant) (Fig. 3c) drastically reduced, but not completely abolished, the interaction with CK1ε as confirmed by coimmunoprecipitation and resulted in the diminished phosphorylation at the CK1ε target site S643 (Fig. 3d–g). Functionally, DVL3 ΔALL failed to activate the downstream Wnt/β-catenin pathway in the TopFlash reporter assay (Fig. 3h) in HEK293 DVL1-2-3$^{-/-}$ cells.

Intriguingly, the DVL3-CK1ε-binding sites show high sequence conservation (Fig. 3b, top) and are almost identical in humans and *Xenopus* (Fig. 3i). This allowed us to analyze the functional consequences of these deletions also in vivo. The activation of the Wnt/β-catenin pathway results in the axis duplication in *Xenopus*, which is a well-defined functional Wnt/β-catenin signaling readout (Fig. 3j, left). First, we injected a dose of mRNA encoding Dvl3 wild type into the marginal zone of the ventral blastomeres of the four-cell stage *Xenopus laevis* embryos to induce double axis formation (Fig. 3j, right). Not surprisingly, the xDvl3 ΔALL variant (lacking aa 338–350, 609–619, and 693–705 in xDvl3) showed dramatically reduced capacity to induce axis duplication both in the presence and absence of exogenous xCK1ε (Fig. 3j, right). Taken together, these data demonstrate that the identified DVL3 regions represent evolutionary conserved bona fide interaction sites for CK1ε, whose deletion abolishes both CK1ε binding and CK1ε-dependent functions of DVL3.

**CK1ε is required for the conformational dynamics of DVL3.** As the DVL3 ΔALL variant is incapable of complete interactions with CK1ε, we further examined the role of CK1ε in the conformational dynamics of DVL3. Using the FlAsH III sensor as a template, we generated and analyzed the ECFP-DVL3 FlAsH III ΔALL variant (Fig. 4a). Conformational dynamics of DVL3 ΔALL was lost but, interestingly, the FRET efficiency for all three conditions was low—suggesting that DVL3 ΔALL remains in the open rather than the closed conformation. To further analyze this phenomenon, we produced CK1ε-deficient (CK1ε$^{-/-}$) HEK293 cells using the CRISPR-Cas9 system (Fig. 4b). These cells (Fig. 4b) failed to respond to Wnt ligands as demonstrated by the lack of phosphorylation of DVL2 and DVL3, and pS1490-LRP6. DVL3 overexpression in CK1ε$^{-/-}$ cells failed to induce Wnt/β-catenin pathway activation monitored by TopFlash in the absence of exogenous CK1ε (Supplementary Fig. 4f). Importantly, the FRET efficiency of the DVL3 FlAsH III sensor in CK1ε$^{-/-}$ cells was low and CK1ε inhibition was unable to increase it as it did in HEK293 wt cells (Fig. 4c). These data suggest that DVL3 in the absence of CK1ε remains in an open (and non-phosphorylated) rather than a closed (and non-phosphorylated) conformation that is observed when CK1ε is present but inhibited by the CK1δ/ε inhibitor

PF670462. One explanation can be non-specific effects of CK1δ/ε inhibitor PF670462, unrelated to CK1ε inhibition. To exclude this possibility, we overexpressed *d*ominant *n*egative (dn) CK1ε mutant P3, which efficiently binds to DVL3 (ref. [33]). The presence of dnCK1ε as well as wild-type CK1ε in combination with PF670462 inhibitor resulted in the significantly increased FRET signal of the DVL3 FlAsH III sensor (Fig. 4d). This confirmed that in presence of inactive CK1ε, DVL3 adopts a more compact conformation which is not the case when CK1ε is absent from the DVL3 complex. In summary with other results, it suggests that CK1ε has a dual role: (i) it retains DVL3 in a compact conformation by physical interaction and (ii) triggers a phosphorylation-induced open conformation of DVL3, when activated.

**C-terminus-PDZ interaction controls conformations of DVL3.** The recent study by Lee et al.[16] has proposed that DVL can switch between open and closed conformations. Using mouse Dvl1 they demonstrated that the closed conformation results from the binding of the conserved terminal 7 aa of the C-terminus (EFFVDIM sequence) to the PDZ domain (for schematics see Fig. 5a). Therefore, we asked whether this interaction contributes at least partially to the DVL3 conformational dynamics detected by our FRET sensor. Indeed, DVL3 lacking the very C-terminal part (the ΔC variant of DVL3 FlAsH III, which lacks aa 699–716; Supplementary Fig. 5b) showed significantly lower basal FRET efficiency than wt DVL3 FlAsH III (Fig. 5b). This suggests that the interaction between the C-terminus and PDZ may play a role in the conformational sampling of DVL.

Our next question was whether CK1-mediated phosphorylation in the PDZ domain and the C-terminus contributes to the conformational dynamics induced by CK1ε. First, we employed the MS/MS approach to analyze the phosphorylation of DVL3 induced by CK1ε in the PDZ domain of DVL3. Two phosphorylation sites highly induced by CK1ε were found in the PDZ domain: S268 and S311 in DVL3 immunoprecipitated from HEK293 cells (Fig. 5c) as well as in the in vitro purified full-length DVL3 phosphorylated by recombinant CK1ε in vitro (Fig. 5d). Furthermore, C-terminal S700 in DVL3 was found to be constitutively phosphorylated in HEK293 cells, as previously mentioned (Supplementary Fig. 4b). All three serine residues are fully conserved (Supplementary Fig. 5a, b). To investigate how the phosphorylation status of S268 and S311 in the PDZ domain influences DVL3 conformation, we created ECFP-DVL3 FlAsH III wt construct that was either non-phosphorylatable (S268A/S311A) or phosphorylation-mimicking (S268E/S311E). DVL3 FlAsH III S268A/S311A variant showed higher FRET efficiency indicating the closed conformation, whilst the FRET efficiency of the DVL3 FlAsH III S268E/S311E variant was lower (Fig. 5e), suggesting predominantly the open conformation for this variant. Intriguingly, the FRET efficiency was not significantly affected by the CK1ε inhibition or CK1ε overexpression in any of these variants (Fig. 5e).

Altogether, these data suggest that the CK1-induced phosphorylation of S268 and S311 can control the conformation of DVL3. This led us to investigate whether this regulation can be explained at the atomic level as a result of weaker binding of the (phosphorylated) C-terminus to the PDZ domain phosphorylated at S268 and/or S311.

**PDZ domain phosphorylation regulates the C-terminus binding.** We performed in silico simulations to investigate the interaction between the C-terminal peptide of human DVL3 (aa 709–716) and human DVL3 PDZ (aa 245–338) domain. We found that the C-terminal peptide binds to the PDZ domain in

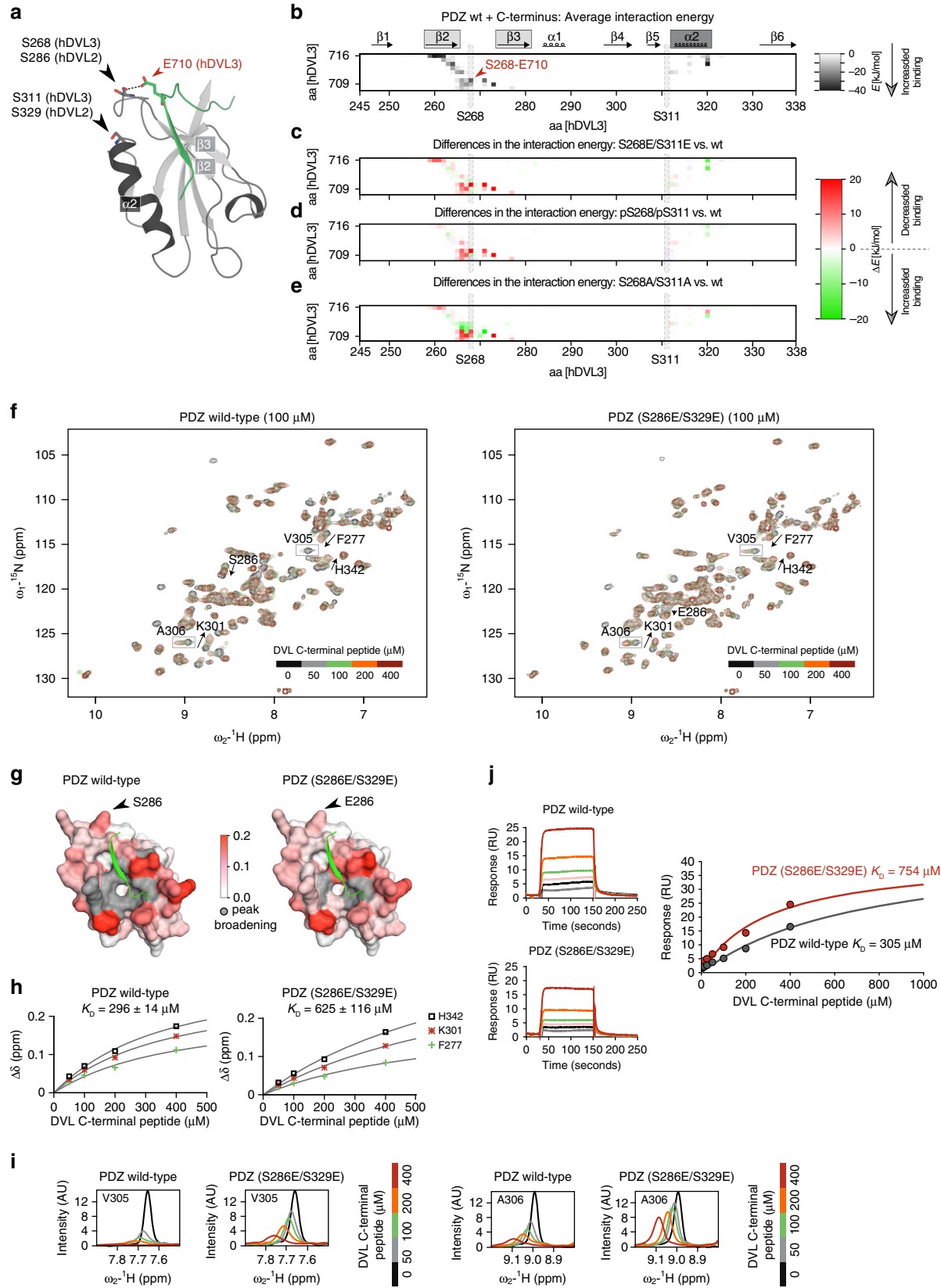

the canonical binding groove (primary binding site[34]) forming an antiparallel β-strand with β2-strand of the PDZ domain (Fig. 6a). Moreover, this interaction extends to the β2–β3 loop, where S268 via its hydroxyl forms a hydrogen bond with the carboxyl group of E710 (Fig. 6b). β2–β3 Loop has been reported as an additional

binding site in the PDZ domains of several other proteins[34–36] but in DVL, such interaction (S268-E710) has not yet been reported.

To estimate the effect of phosphorylation on peptide binding in silico, we investigated the interaction between the peptide and the

**Fig. 6** PDZ phosphorylation regulates the interaction with the DVL C-terminal peptide. **a** Structure of DVL3 C-terminal peptide (aa 702–716; green) bound to the PDZ-binding pocket (PDZ domain from DVL3, aa 245–338; in silico α-helices in dark and β-strands in light gray) observed in the in silico simulations. S268, S311, and C-terminus residue E710 participating in hydrogen bonds (dotted line) are highlighted with a stick model. **b** The matrix of mean interaction energy between each residue of PDZ domain aa 245–338 (x-axis) and DVL C-terminal peptide aa 709–716 (y-axis). Strength of the attraction of PDZ wt and C-terminus is depicted in white-black gradient. **c–e** The differences in mean interaction energies shown as a difference from wild-type PDZ wt: **c** PDZ (S268E/S311E), **d** PDZ (phospho-S268/phospho-S311), and **e** PDZ (S268A/S311A) between PDZ mutants. The change in the interactions are depicted in green (stronger) or red (weaker). Interactions of protein/peptide end caps are also displayed in the matrix. **f–i** NMR titrations of the DVL2 PDZ wild type and PDZ phosphomimicking variant (S286E/S329E; corresponding to S268 and S311 in DVL3) with DVL C-terminal peptide. **f** Overlay of $^{1}$H, $^{15}$N HSQC spectra for each titration point. Arrows indicate selected residues that exhibit fast exchange properties on the NMR timescale and gray boxes selected residues that exhibit intermediate-to-fast exchange properties on the NMR timescale. **g** Mapping of chemical shift perturbations on DVL-peptide (in green) structure from PDB database (PDB ID: 3CCO). Fast exchange residues colored using a gradient from white to red according to chemical shift perturbation and intermediate exchange residues that go beyond detection at the end of the titration colored in gray. S286, or E286 substitution, of DVL2 are highlighted by black arrow. **h** Binding isotherms for three residues that experience fast exchange during NMR titration. The apparent $K_D$ values represent the mean with the standard deviation for the three cross-peaks analyzed. **i** Experimental line shapes during the titration for two selected residues that experience intermediate-to-fast exchange. **j** SPR sensograms of DVL C-terminal peptide binding to wild type and S286E/S329E PDZ from DVL2 and the corresponding binding isotherms fitted to a one-site binding model. RU stands for response units, AU for arbitrary units

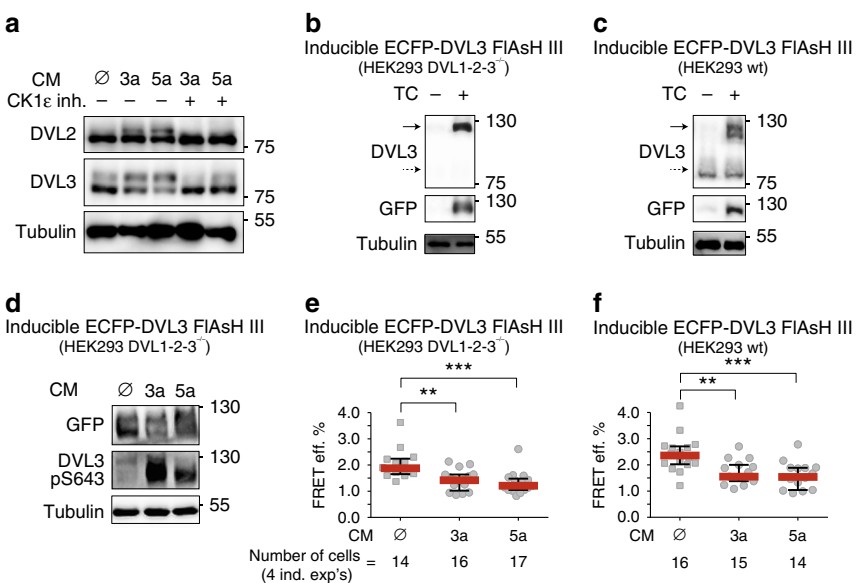

**Fig. 7** Wnt ligands promote open conformation of DVL3. **a** Western blot analysis of the effect of the Wnt-3a (3a) and Wnt-5a (5a) ligands on the phosphorylation of endogenous DVL2 and DVL3 in HEK293 wt cells. **b, c** Western blot analyses of samples from two stable cell lines (derived from HEK293 wt and HEK293 DVL1/2/3 triple knockout[22] cells) that inducibly under tetracycline-controlled promoter express ECFP-DVL3 FlAsH III sensor. Dashed arrow indicates endogenous DVL3; full arrow indicates ECFP-DVL3. **d** Western blot analysis of the effect of the Wnt-3a (3a) and Wnt-5a (5a) ligands on the phosphorylation status of ECFP-DVL3 FlAsH III in HEK293 DVL1/2/3 triple knockout[22] cells. **e, f** Measurements of the intramolecular FRET efficiency of the endogenously expressed DVL3 FlAsH sensors III after the treatment with Wnt ligands in HEK293 wt and HEK293 DVL1/2/3 triple knockout[22] cells. One data point corresponds to one analyzed cell; datapoints from four independent transfections were merged. Data in **e** and **f** represent median ± interquartile range. Statistical significance in **e** and **f** was analyzed by one-way ANOVA test with Gaussian distribution and Tukey's post-test (*, $p \leq 0.05$; **, $p \leq 0.01$; ***, $p \leq 0.001$, ****, $p \leq 0.0001$; ns, not significant, $p > 0.05$). CM conditioned medium, Inh. inhibitor, TC tetracycline

DVL3 PDZ with two phosphorylated serines (pS268/pS311), phosphomimic S268E/S311E, and non-phosphorylatable S268A/S311A. In all cases, the peptide remained bound to the PDZ during the entire simulation (500 ns). However, the enthalpic interaction between the PDZ and the peptide were weaker for both phosphorylated and the mimetic variant by roughly 20% when compared to PDZ wt (Fig. 6c, d). In contrast, the alanine double mutant showed similar interaction strength as wt (Fig. 6e). The changes to the interaction strength were mostly situated in proximity to mutated/phosphorylated sites in the β2–β3 loop with a minor change in the primary carboxyl binding site (see Fig. 6b). The observed interaction strengths correlate with the preference of each PDZ mutant for open or closed conformation observed in the in vivo experiments and provide a candidate mechanism supporting FRET experimental observations.

In order to address the possible role of DVL3 phosphorylation-driven regulation involving S268 and S311 in vitro, we performed NMR structural characterization (Fig. 6f–i). For this purpose, we exploited the well-characterized PDZ domain of human DVL2 (aa 265–361)[14], where S286 and S329 correspond to S268 and S311 of human DVL3. Investigation of chemical shift differences revealed that the phosphorylation-mimicking substitutions (S286E/S329E) do not affect the PDZ fold[16] (Supplementary Fig. 6a–c). Next, we performed NMR titrations with a phosphorylated (pS700) C-terminal peptide ligand derived from DVL3 (aa 698–716). The phosphorylation variant was chosen as the most physiological, since S700 appeared as constitutively phosphorylated (see Supplementary Fig. 4b). The peak trajectories induced by peptide binding are linear and follow the same course for both PDZ wt and phosphorylation-mimicking variant

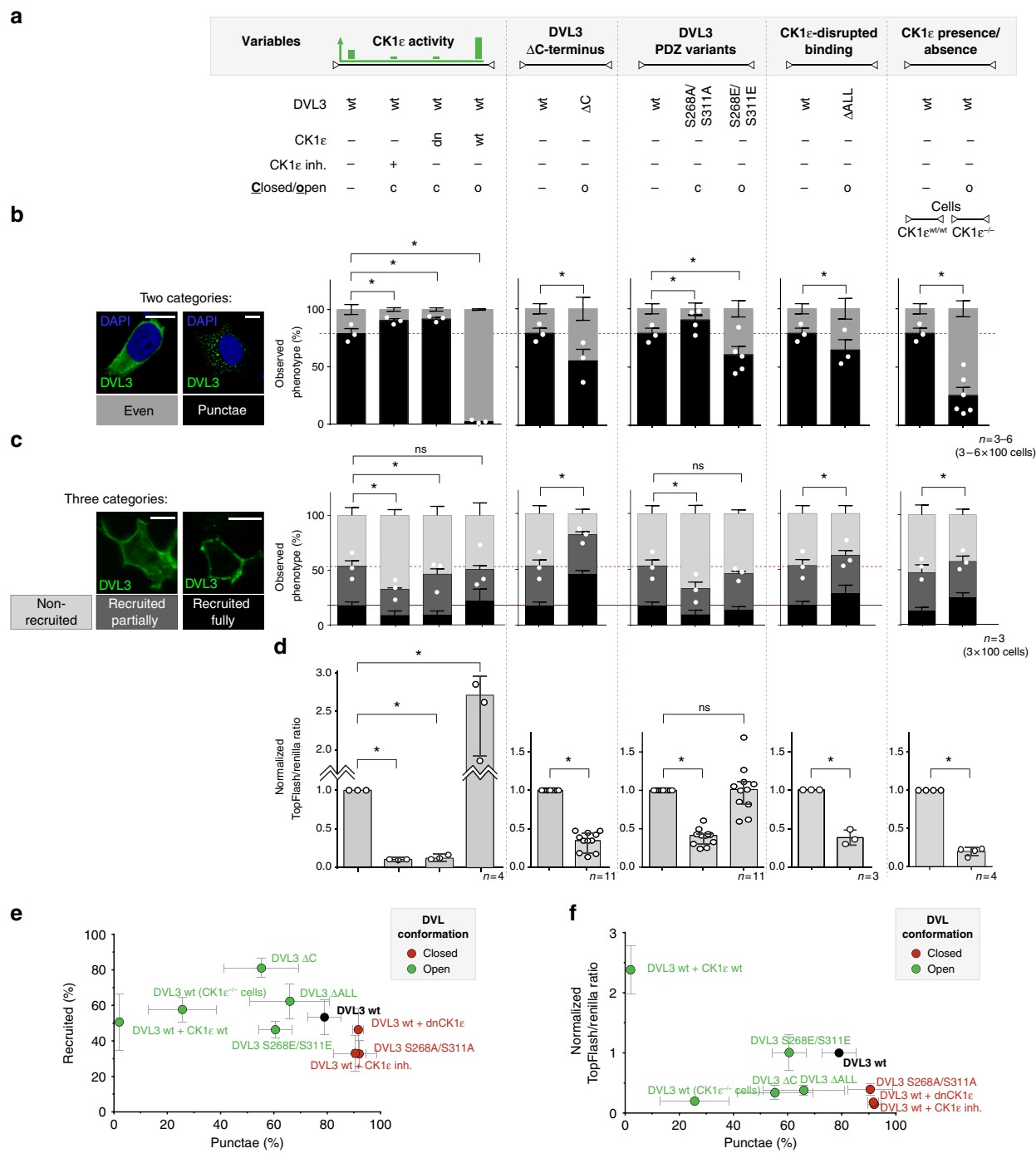

indicating a similar binding mode. Of the two serine residues, only S286 is affected by peptide binding and clearly experiences larger perturbation than its glutamic acid counterpart (Fig. 6f, Supplementary Fig. 6d). This is in line with in silico data, proposing that S286 in DVL2 (corresponding to S268 in DVL3) is directly involved in contacts with the C-terminal peptide.

Intermediate and intermediate-to-fast exchange peaks correspond to PDZ residues of the binding groove (Fig. 6g). These residues experience severe line broadening throughout the course of the ligand titration in PDZ wt when compared to S286E/S329E variant (Fig. 6i). Fast exchange peaks correspond to residues at the groove periphery. Their quantitative analysis shows that the phosphorylation-mimicking variant has a two-fold weaker

affinity for the peptide (Fig. 6h). In principle, the chemical shift perturbations are larger for PDZ wt, indicating that the equilibrium is shifted further to the bound state when compared to the PDZ phosphorylation-mimicking variant (Supplementary Fig. 6d). As an independent confirmation of this conclusion, we also performed surface plasmon resonance (SPR) binding experiments (Fig. 6j). The apparent dissociation constants derived from SPR are in line with the NMR observations and reported a two-fold lower peptide affinity for the PDZ phosphorylation-mimicking variant.

In summary, the biophysical data and the in silico simulations indicate a phosphorylation-driven mechanism that modulates the binding of DVL C-terminus to PDZ domain. In combination

**Fig. 8** DVL conformations correlate with DVL subcellular localization and membrane recruitment. **a** Conditions and variants with the defined conformational state identified by FlAsH-based FRET subjected to analyses in **b**–**d**; o indicates open, c indicates closed. **b** ECFP-tagged DVL3 was transfected to HEK293 wt cells as indicated. Subcellular localization was analyzed by anti-GFP immunostaining. Based on the pattern of ECFP-DVL3, cells were classified to have either even or punctate localization (plotted as white dots) of DVL3. Typical examples are shown in the left. Data represent average from three independent transfections with 100 cells counted in each condition per transfection. Identical control condition is shown several times for better clarity. **c** FLAG-tagged DVL3 and FZD6-mCherry were transfected in HEK293 DVL1/2/3$^{-/-}$, wt, and CK1ε$^{-/-}$ cells as indicated. Subcellular localization was analyzed by anti-FLAG immunostaining. Based on the membrane localization of FLAG-DVL3, cells were classified to possess DVL3 either non-recruited, partially recruited, or fully recruited (the sum of partial and full DVL3 recruitment plotted as white dots). Typical examples are shown in the left. Data represent average from three independent transfections with 100 FZD6-mCherry-positive cells counted in each condition per transfection. Identical control condition is shown several times for better clarity. **d** Analysis of the Wnt/β-catenin downstream signaling monitored by Dual Luciferase TopFlash/Renilla Reporter Assay in HEK293 cells. Data in **b**–**d** represent mean ± S.D. and statistical significance in **d** was analyzed by one-way ANOVA test with Gaussian distribution and Tukey's post-test. For more details about the statistics used in Fig. 8b and c, please see the appropriate section in Methods (*, $p \leq 0.05$; ns, not significant, $p > 0.05$). **e**, **f** Data obtained in **b**–**d** were plotted in the 2D graphs. Red color indicates the closed conformation; green indicates open. Error bars show S.D. for each parameter. % of recruited is a sum of partially recruited and fully recruited in **c**. DVL variants that are closed form a distinct population when membrane recruitment and subcellular localization is considered (**e**) but not when the TopFlash assay is included (**f**, and Supplementary Fig. 7a); dn dominant negative

with other results in this study, we propose that such mechanism contributes to the CK1-dependent control of DVL conformational changes.

**Wnt ligands promote open conformation of DVL3**. It has been shown previously that stimulation with Wnt ligands activates CK1δ/ε that in turn phosphorylate DVL. Wnts from both major groups—i.e. capable to activate Wnt/β-catenin (i.e. Wnt-3a) or alternative non-canonical Wnt pathway (i.e. Wnt-5a)—trigger phosphorylation of DVL that is dependent on CK1δ/ε activity (Fig. 7a). Thus, we decided to test if/how Wnts change the conformation of DVL3. To do this, we generated two stable cell lines (derived from HEK293 wt and HEK293 DVL1/2/3 triple knock-out[22] cells) utilizing tetracycline-controlled promoter driving expression of ECFP-DVL3 FlAsH III sensor (Fig. 7b, c). Under these conditions, the ECFP-DVL3 FlAsH III gets expressed at levels comparable to endogenous DVL3 (Fig. 7c), and in response to Wnt stimulation gets phosphorylated—this can be monitored either as a phosphorylation-dependent shift or by anti-phospho-S643 antibody (Fig. 7d). Analysis of FRET efficiency in both cell lines showed that basal FRET efficiency (≈2%) was comparable with the control conditions analyzed upon transient over-expression of ECFP-DVL3 FlAsH III (see Figs. 2 and 4). Importantly, treatment with both Wnt-3a and Wnt-5a resulted in significantly lower FRET, regardless of the cell line used (Fig. 7e, f). Altogether, these results show that stimulation with Wnt ligands changes DVL3 conformation similarly to the over-expression of CK1ε and suggest that the phosphorylation-driven conformational switch participates in the mechanism of Wnt-induced DVL activation.

**Open DVL3 is evenly localized and better membrane recruited**. Our analyses of DVL3 conformational dynamics by FRET identified multiple conditions that promote either open or closed conformations of DVL3 (see Figs 2, 4 and 5). These conditions/mutations (schematized in Fig. 8a) include changes in the CK1ε activity, presence/absence of DVL3 C-terminus, phosphomimicking/blocking mutations in the PDZ domain, capacity of DVL3 to interact with CK1ε, and presence or absence of CK1ε in cells. Altogether, we could identify five cases associated with open conformation and three cases associated with closed conformation. In order to link DVL3 conformational changes with its biological properties, we decided to perform a systematic correlative analysis. The following conditions were included as closed: DVL3 wt + CK1δ/ε inhibitor PF-670642, DVL3 wt + dnCK1ε, and DVL3 (S268A/S311A), or open: DVL3 wt + CK1ε wt, DVL3

ΔALL, DVL3 (S268E/S311E), DVL3 ΔC, and DVL3 wt in CK1ε$^{-/-}$ cells.

First, we analyzed the preferred subcellular localization of DVL3 (as introduced in Fig. 2c)[10,14]. As shown in Fig. 8b, the open variants of DVL3 are significantly more evenly distributed whereas closed DVL3 is more punctate. Next, we investigated if DVL3 conformation correlates with the capacity of DVL3 to be recruited by FZD6, a key Wnt receptor, to the plasma membrane. For this purpose, we co-expressed FZD6 and analyzed membrane recruitment of DVL3 in FZD6-positive cells by immunofluorescence[37]. Individual cells were categorized as: not recruited (Fig. 8b, no DVL3 in the membrane), partially recruited (clear membrane as well as cytoplasmic localization), and fully recruited (predominantly plasma membrane localization, Fig. 8c, left). This analysis showed that closed DVL3 is less efficiently recruited to the membrane by FZD6, whereas open DVL3 behaves either as a control or is recruited more efficiently (Fig. 8c). In the last readout we analyzed the capacity of DVL3 to activate down-stream Wnt/β-catenin pathway monitored by TopFlash reporter assay (Fig. 8d). Under all conditions where DVL3 is closed—DVL3 (S268A/S311A), DVL3 wt + CK1δ/ε inhibitor, and DVL3 wt + dnCK1ε—no (or negligible) TopFlash/Renilla activation was observed. On the other hand, under conditions that favor the open conformation both activation as well as inhibition of the Wnt/β-catenin pathway was evident.

Next, we plotted the biological properties of individual open and closed cases in a single graph (Fig. 8e, f; Supplementary Fig. 7a). When both subcellular localization and FZD6-induced membrane recruitment were considered, open and closed conditions separated from wt DVL3 very clearly (Fig. 8e). Ability to activate TopFlash was observed only when CK1ε was co-expressed with wt DVL3. Thus, it seems that CK1ε activity represents an independent additional factor that controls the behavior of open DVL3 forms; i.e., open conformation is not sufficient (although it can be required) for DVL3 to activate downstream Wnt/β-catenin signaling. Altogether these data demonstrate a strong correlation between DVL3 conformations and their ability to be recruited to the receptor complex (FZD6) or localize evenly in a cell.

In the cell culture model we were unable to dissect the effects on the non-canonical Wnt/PCP pathway, and as such we turned into the *Xenopus* embryo model. Alterations in the Wnt/PCP pathway activity result in the convergent extension (CE) defects (Supplementary Fig. 7b, right). In order to avoid any artifacts, we tested the constitutively open and closed variants of xDvl3 based on point mutations or small deletions—namely open xDvl3 ΔC and xDvl3 (S267E/S310E) and closedxDvl3 (S267A/S310A).

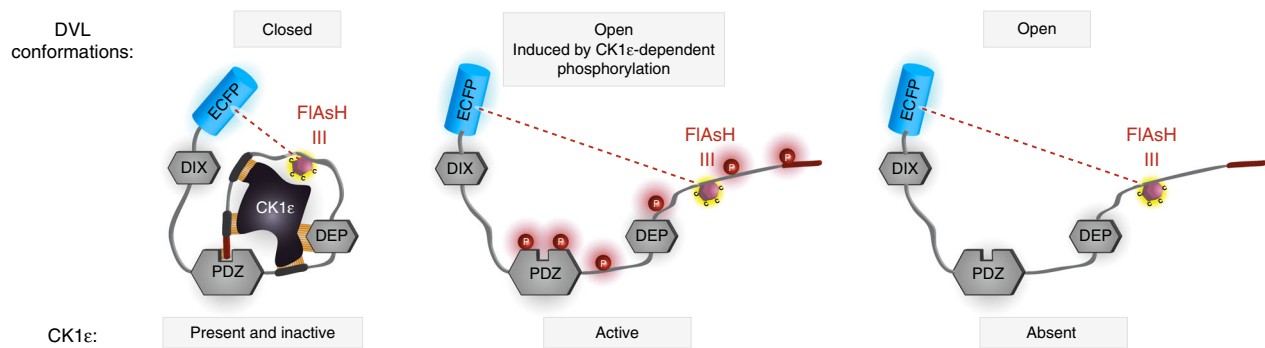

**Fig. 9** Summary showing CK1ε role in DVL3 conformational dynamics. A summarizing model which proposes at least three DVL conformations in vivo: (i) a closed (CK1ε present and inactive), (ii) open (CK1ε active), and (iii) non-physiological open, which occurs when CK1ε is absent or the DVL-CK1ε interaction is disrupted. Position of insertion of FlAsH III binding tag is indicated. The CK1-induced phosphorylation events are depicted as P in red circle and the C-terminus of DVL as red thick line. The molecular distance analyzed in the FRET FlAsH sensor III is shown as a dashed red line; ECFP, enhanced cyan fluorescent protein

Phosphorylation sites in the PDZ domain are fully conserved between human and Xenopus (for alignment see Supplementary Fig. 5a) and hDVL3 S268/S311 corresponds to xDvl3 S267/S310. As shown in Supplementary Fig. 7b, open variants of xDvl3 showed similar potency to induce CE defects as wt xDvl3 with approximately 70–80% embryos displaying CE phenotype, whereas the closed xDvl3 (S267A/S310A) showed significantly reduced capacity to induce CE/NT (neural tube) defects with less than 50% affected embryos. These results correlate strongly with the capacity of the corresponding human DVL3 variants to be recruited to the membrane by FZD6 (Fig. 8c).

## Discussion

In this study we establish an FlAsH-based single-cell FRET tool for the investigation of DVL conformations in intact living cells. Using this approach, we identify a key role of CK1ε, a major DVL kinase, in controlling DVL conformation in vivo.

The concept of DVL conformational sampling was first introduced by Jung et al.[38]. They proposed that the C-terminus of DVL is a ubiquitin-binding domain that can recognize K63-linked ubiquitin chains in the DEP domain to yield an inactive, closed, conformation[38]. Subsequent studies of Zheng group[16,17], in vitro proposed an alternative open/closed conformational switch mediated by docking of the C-terminus to the PDZ domain of DVL. The FlAsH-based FRET DVL3 sensor technique allows us to identify a key role of phosphorylation in the regulation of this process. Specifically, we identify two CK1ε-induced phosphoserine residues (S268 and S311 in human DVL3) in the PDZ domain whose mutation dramatically interferes with the DVL3 conformational sampling, and further supports the C-terminus–PDZ interaction model[16,17].

Previously, we have shown[28] that efficient interaction of DVL3 and CK1ε requires multiple DVL3 regions. The DIX domain is the only region of DVL3 which seems to be dispensable for high affinity interaction[28]. The DEP domain appears to be the most important structured domain involved in CK1ε binding[28,39]; here we identify three regions in the IDRs of DVL3 that are involved in CK1ε interactions. DVL3 lacking these three short sequences is not only inefficient to bind CK1ε but it is also not amenable to conformational rearrangements. Interestingly, such DVL3 variant is found counterintuitively in the open conformation. The same applies when CK1ε is eliminated in the CK1ε$^{-/-}$ cell line. This suggests that CK1ε can also have an important scaffolding function required to retain DVL3 in the closed conformation. This view is schematized in Fig. 9. Similar function has been observed for other kinases that regulate biological processes not

only by their catalytic activity but also by their scaffolding function[40]. In principle, the requirement for CK1 as a scaffold can help to elucidate earlier findings shown by genetic experiments in *Drosophila* that Wnt/PCP pathway requires CK1ε but not its kinase activity (in contrast to Wnt/β-catenin pathway that requires both)[41].

The key question is how DVL conformation affects the formation of Wnt signaling complexes. We can show that both Wnt-3a, which triggers Wnt/β-catenin pathway, and Wnt-5a ligand, which activates non-canonical Wnt pathway, promote open conformation of DVL3. Both these ligands activate CK1ε, which results in a very similar or identical phosphorylation of DVL[24]. At the same time, we could correlate open conformations of DVL3 with the efficient FZD6-dependent membrane recruitment of DVL3. A similar correlation was reported recently using XDsh, XFz7, and zebrafish embryonal cells[17]. Based on these facts, we propose a model where Wnt-induced activation of CK1ε leads to the phosphorylation of S268/S311 in the PDZ domain and an open DVL conformation; i.e. open DVL3 then gets more efficiently bound to FZD. The molecular mechanism behind the more efficient recruitment to FZD-centered receptor complex is unknown but several candidate mechanisms reported in the literature can be considered for further testing. First, it is possible that in a closed conformation (i.e. when the binding groove of PDZ is occupied by DVL C-terminus), a conserved KTXXXW motif in the C-termini of FZDs that is capable of direct interaction with the PDZ[42,43] domain cannot interact with DVL. Second, the liberation of the DVL C-terminus in the open conformation provides a binding interface that can be recognized by proteins that facilitate receptor complex assembly. Such a candidate In the non-canonical Wnt pathway could be Ror2, which binds the C-terminus of DVL and more efficiently recognizes DVL phosphorylated by CK1 (ref. [44]). Third, our data show that the closed variants of DVL3 are more punctate whereas open variants are more evenly distributed. These phenotypes approximate the multimerization properties of DVL mediated by DIX and DEP domains[10,14]. In all our conditions, DVL3 has intact DIX and DEP domains, which suggests that the conformation itself affects multimerization and/or biophysical properties of DVL. One can speculate that closed DVL3 represents an inactive and immobile pool of DVL whereas open DVL is more mobile. Existing reservoirs of inactive or autoinhibited DVL have been proposed before[17,45]—for example: DVL-binding protein nucleoredoxin (NXN) can recognize such a pool and prevent it from degradation[45,46]. The key function of CK1 upon activation of Wnt pathway can then be mobilization of DVL from the inactive pool to be followed by FZD-mediated events—this distinction is

in line with earlier findings showing that the action of CK1ε and FZD on DVL is molecularly distinct[29].

Taken together, we provide evidence that Wnts and CK1ε regulate the conformation of DVL3, and that DVL3 in the open conformation, induced by these stimuli, is more mobile and more efficiently recruited to FZD. The FlAsH FRET DVL3 sensor technique provides the tool for the follow-up studies that can shed light on the diverse DVL functions at the membrane, in the centrosome, in the nucleus, and mainly on the regulation of DVL function in the Wnt/β-catenin and Wnt/PCP signaling pathways.

## Methods

**Cell culture and transfection and mutagenesis and treatments.** HEK293 wild type (ATCC-CRL-11268), DVL1-2-3$^{-/-}$ (published in ref. [22]), ECFP-DVL3 FIII and CK1ε$^{-/-}$ cell lines (both derived from the aforementioned HEK293 wild-type cells) were grown at 37 °C and 5% (vol/vol) CO$_2$ in Dulbecco's modified Eagle's medium (DMEM; Gibco #41966-029), 10% (vol/vol) fetal bovine serum (Gibco, #10270), 2 mM L-glutamine (Life Technologies, #25030024), and 1% (vol/vol) antibiotics (penicillin/streptomycin; Hyclone-Biotech, #SV30010).

For immunofluorescence, cells were seeded (40,000 per well) on 24-well plates with gelatin-coated coverslips and transfected, as indicated, with 0.1 µg of each corresponding plasmid (0.3 µg of mCherry-FZD6) for 6 h. For western blotting and Dual Luciferase TopFlash/Renilla Reporter Assay, cells were seeded (60,000 per well) directly on 24-well plates with 0.1 µg of each corresponding plasmid (if not indicated otherwise) for 6 h. For coimmunoprecipitation, cells were seeded (4.4 × 10$^6$) on 10-cm dishes and transfected with 1 µg of each corresponding plasmid. The next day after seeding, cells were transfected using polyethyleneimine (PEI) in a stoichiometry of 6 µl of PEI per 1 µg of DNA for 6 h and were harvested 24 h after transfection. The following plasmids used in HEK293 cells have been described previously: FLAG-DVL3 wt[47], CK1ε wt[48], mCherry-FZD6 (ref. [37]), and dominant-negative CK1ε P3 (ref. [33]). The ECFP-DVL3 wt plasmid was made by Gateway Technology (via LR recombination according to the manufacturer's instructions; Thermo Fisher Scientific, #11791020) from the donor plasmid pDONR221 DVL3 (DNASU, #FLH178665.01x) and the destination plasmid pdECFP (LMBP, #4548). The HA-DVL3 wt plasmid was made by Gateway Technology from the donor plasmid pDONR221 DVL3 (DNASU, #FLH178665.01x) and the destination plasmid pcDNA3.1 N-term HA (gift from Erich Nigg).

Mutagenesis reactions were performed using the QuikChange II XL Site-Directed Mutagenesis Kit following the manufacturer's instructions (Agilent Technologies, #200521). All mutations described in this study were verified by Sanger sequencing. All primers used for the site-directed mutagenesis are listed in Supplementary Table 1. All plasmids and more detailed information are available upon request.

Cells were treated as indicated by 10 µM CK1δ/ε inhibitor PF-670642 (Santa Cruz, #sc-204180A) overnight; 1 µM LGK974 Porcupine inhibitor (Stem RD, #974-02) overnight; 1 µg/ml of tetracycline for 4 h; and by Conditional Media (CM) for 3 h (WB, FlAsH-based FRET approach) or 8 h (TopFlash/Renilla Dual luciferase assay), which were obtained from the control or Wnt-3a/Wnt-5a stably overexpressing L-cells mouse cell line (L-Wnt3a; ATCC-CRL-2647 and L426 Wnt5a cells; ATCC-CRL-2814) according to the ATCC instructions.

**CRISPR-Cas9 generation of HEK T-REx-293 CK1ε$^{-/-}$ cell line.** The sequence of gRNA targeting CSNK1E gene was chosen according to Broad Institute CRISPRko designing tool as follows: gRNA (CSNK1E)—AAGTTCTACAAGATGATGCA (GGG) (SfaNI restriction site indicated, PAM sequence in brackets).

The following plasmids were used for gRNA cloning: pSpCas9(BB)-2A-GFP (PX458) (Cas9 from *S. pyogenes* with linked EGFP and cloning backbone for gRNA, Addgene plasmid #48138[49] and pU6-(BbsI)_CBh-Cas9-T2A-mCherry (Cas9 from *S. pyogenes* with linked mCherry and cloning backbone for gRNA, Addgene plasmid #64324[50].

T-REx-293 cells (Invitrogen) were cultured according to the manufacturer's instructions and transfected by Lipofectamine® 2000 DNA Transfection Reagent (Thermo Fisher Scientific) with #48138 (cloned in gRNA for CSNK1E). Cells were subsequently subjected to FACS (fluorescence activated cell sorting) where simultaneously GFP (green fluorescent protein) and mCherry-positive cells were seeded in 96-well plates in one cell per well format.

Monoclonal cell lines were backed up and subsequently screened by western blot using CK1ε antibody following the CK1ε antibody (Santa Cruz, #sc-6471). For verification and specification of modification were above-mentioned PCR products cloned into pcDNA3 plasmid (Invitrogen) and sequenced using CMV forward primer: CGCAAATGGGCGGTAGGCGTG.

**Generation of HEK T-REx-293 ECFP-DVL3 FlAsH III cell lines.** cDNA encoding ECFP-DVL3 FlAsH III was amplified by Pfx50 DNA Polymerase (Thermo Fisher Scientific, #12355012) using primers containing sequences recognized by *Hin*dIII (forward primer TATAAGCTTatggtgagcaaggg; *Hin*dIII site underlined) and *Xba*I (reversed primer AATTCTAGAtcacatcacatcccacaaa; *Xba*I site underlined)

restriction enzymes (Thermo Fisher Scientific, #ER0501 and #ER0681), extracted from gel using GeneJET Gel Extraction Kit (Thermo Fisher Scientific, #K0691) and ligated into *Hin*dIII- and *Xba*I-linearized pcDNA4/TO (Thermo Fisher Scientific, #K102001) vector using T4 DNA Ligase (New England Biolabs, #M0202) according to the manufacturer's protocol. Then competent *E. coli* DH5α bacteria were transformed with ligation mix using standard heat-shock protocol, individual bacterial colonies were picked, cultured, and plasmid DNA was isolated by GenElute Plasmid DNA Miniprep Kit (Sigma-Aldrich, #PLN70). Plasmids were checked by Sanger sequencing (forward sequencing primer ATTGACG-CAAATGGGCG; reverse GCCTTCCTTGACCCTGGA) for the presence of the ECFP-DVL3 FlAsH III wt cDNA. Next, T-REx-293 cells (Invitrogen) and their derivate HEK293 DVL1-2-3$^{-/-}$ cell line[22] were transfected with pcDNA-4-TO-ECFP-DVL3 plasmid using PEI. Twenty-four hour post-transfection culture medium was changed for the new one supplemented with 200 µg/ml for generation of stable cell lines. After selection clonal lines were obtained using the serial dilution method and analyzed by western blot using the GFP/DVL3 antibodies for the inducible transgene expression by treatment with 1 µg/ml of tetracycline.

**Dual luciferase TopFlash/Renilla Reporter Assay.** Cells were transfected 24 h after seeding with 0.1 µg of Super8X TopFlash construct, 0.1 µg of Renilla luciferase construct, and 0.1 µg of corresponding plasmids per well for 6 h in a 24-well plate. As for the ECFP-DVL3 wt stable cell lines, transfection was followed by 4 h induction of tetracycline (1 µg/ml) and 8 h treatment of CM:DMEM (1:1 ratio). For performing this assay, Promega Dual Luciferase assay kit (Promega, #E1910) was used according to the manufacturer's instructions. Luminescence was measured by a Hidex Bioscan Plate Chameleon Luminometer. Results are depicted as the ratio of TopFlash and Renilla signal (TopFlash fold induction), which was normalized for each experiment to certain control column (normalized TopFlash fold induction). Data were analyzed by MS Excel 2007 and GraphPad Prism 6 and results were shown as the means ± S.D. (number of experiments indicated ad hoc).

**Western blot and sample preparation.** Coimmunoprecipitation protocol was used modified from Bryja et al.[29] DMEM was removed after 24 h, cells were washed by phosphate-buffered saline (PBS) and 1 ml of cold NP40-based lysis buffer (50 mM Tris buffer, pH 7.4; 300 mM NaCl; 1 mM EDTA; 0.5 % NP40) supplemented with 1× protease inhibitors (Roche Applied Science, #11836145001), 0.1 mM DTT and 1× phosphatase inhibitors (Calbiochem, #524625) was used per one 10-cm dish. The lysate was collected after 15 min of lysis on 4 °C and was cleared by centrifugation at 16.1 r.c.f. for 30 min. Samples with the antibody (1.25 µg of antibody per sample) were incubated on carousel overnight. In the morning, 30 µL of G protein-Sepharose beads (GE Healthcare, #17-0618-05)—previously blocked overnight in NP40-based lysis buffer supplemented with 5% bovine serum albumin (BSA) and in the morning equilibrated in BSA-free complete NP40-based lysis buffer—were then added to samples. After 4 h of incubation on carousel, samples were washed four times, 40 µL of 2× Laemmli buffer were added, and samples were boiled. The antibody used for immunoprecipitation was anti-CK1ε (BD Biosciences, #610445).

Blot and sample preparation were performed as previously described[51] and developed using chemiluminescence documentation system FusionSL (Vilber-Lourmat). The antibodies used were: anti-FLAG M2 (1:1000; Sigma-Aldrich, #F1804), anti-FLAG (1:1000; Sigma-Aldrich, #F7425), anti-DVL3 (1:1000; Santa Cruz, #sc-8027), anti-DVL3 (1:1000; Cell Signaling, #cs-3218S), anti-DVL2 (1:1000; Cell Signaling, #cs-3216S), anti-CK1ε (1:500; BD Biosciences, #610445), anti-CK1ε (1:500; Santa Cruz, #sc-6471), anti-GFP (1:2000; Fitzgerald, #20R-GR-011), anti-LRP6 pS1490 (1:250; Cell Signaling, #cs-2568), anti-tubulin (1:2000; Sigma-Aldrich, #T6199), anti-DVL3 phospho S643 (ref. [29]) (1:1000), and anti-DVL3 phospho S697 (ref. [52]) (1:1000). All relevant uncropped blots are available in the Source Data file.

**Immunofluorescence.** Cells were seeded on gelatin-coated coverslips in 24-well plates and were transfected the next day. Cells were fixed 24 h later in fresh 4% paraformaldehyde, permeabilized with 0.5% Triton X-100, blocked with PBS/BSA/Triton/Azide buffer (PBTA) [3% (wt/vol) BSA, 0.25% Triton, 0.01% NaN$_3$] for 1 h, and incubated overnight with primary antibodies (in dilution 1:500) in PBTA at 4 °C. The next day, the coverslips were washed in PBS and incubated with secondary antibodies conjugated to Alexa Fluor 488 (Invitrogen A11001) or/and Alexa Fluor 594 (Invitrogen A11058) (all in dilution 1:1000), washed with PBS [in case of samples containing FLAG-DVL3 WT and derived variants stained with DAPI (1:5000)], and all coverslips were mounted on microscopic slides. Cells were then visualized on an Olympus IX51 fluorescent microscope using ×40 air or ×100 oil objectives and/or an Olympus Fluoview 500 confocal laser scanning microscope IX71 using ×100 oil objective. One hundred positive cells per experiment (n = 3) were analyzed and scored according to their phenotype into two/three categories (punctae/even); or non-recruited/partially/fully recruited for FZD6-dependent recruitment of DVL3. The antibodies used were as follows: anti-FLAG M2 (Sigma-Aldrich, #F1804), anti-DVL3 (Santa Cruz, #sc-8027), anti-CK1ε (Santa Cruz, #sc-6471), anti-RFP (Chromotek, #5h9) and anti-GFP (Fitzgerald, #20R-GR-011).

**Xenopus laevis embryos.** The work with *Xenopus laevis* was carried out according to the German animal use and care law (Das Tierschutzgesetz) and approved by the local authorities and committees (animal care and housing approval: I/39/EE006, Veterinäramt Erlangen; animal experiments approval: 54-2532.2-8/08, German state administration Bavaria/Regierung von Mittelfranken). The generation and cultivation of *Xenopus* embryos was done in accordance with general protocols and staged after the normal table of Niewkoop and Faber (1994)[53].

RNA for microinjection was synthesized from plasmids pCS2 myc-xDvl3 (ref. [54]), pCS2 myc-xDvl3 ΔALL (lacking sequence corresponding to aa 338–350, 609–619, and 693–705 in xDvl3), pCS2 xDvl3 (S267A/S310A), pCS2 xDvl3 (S267E/S310E), pCS2 xDvl3 ΔC, and pCS2 xCK1ε[55], respectively, using the mMessage mMachine Kit (Ambion). Embryos were injected at the four-cell stage either into the two ventral blastomeres for testing the constructs' ability to induce a secondary body axis. Injection amounts of synthetic RNA were 500 pg for myc-xDvl3 wild type and the myc-xDvl3 ΔALL variant, and 150 pg for xCK1ε RNA. For the analysis of morphogenetic defects, four-cell stage embryos were injected into the two dorsal blastomeres with 100 pg myc-xDvl3 RNA and the corresponding variant RNAs. After injection, the embryos were cultivated until they reached stage 26, fixed, and analyzed for axis duplication or morphogenetic defects, respectively.

**FlAsH-based FRET.** HEK293 cells were seeded onto round 24-mm coverslips, which were previously placed in six-well plates and coated with 200 μl of poly-D-lysine (1 mg/ml) for 20 min. Cells were transfected 16–18 h later using Effectene (Qiagen), according to the manufacturer's instructions. DMEM was replaced 6 h later and the analysis was done 24 h after transfection. As for the ECFP-DVL3 FIII stable cell lines: next day after seeding, protein expression was induced by 4 h treatment of tetracycline (1 μg/ml), which was followed by 3 h treatment of CM: DMEM (1:1 ratio).

FlAsH labeling of the DVL3 FlAsH sensors was performed as previously described[18,19]. In summary, transfected cells were washed once with Hank's Balanced Salt Solution (HBSS) containing 1.8 g/l glucose and then incubated at 37 °C for 1 h with HBSS supplemented with 500 nM FlAsH; 12.5 μM 1,2-ethanedithiol (EDT); and corresponding inhibitors. In order to reduce non-specific labeling, cells were then rinsed once with HBSS and incubated for 10 min with HBSS containing 250 μM EDT and corresponding inhibitors. Last, cells were washed twice with HBSS and maintained in DMEM medium with corresponding inhibitors (for the ECFP stable cells —CM:DMEM (1:1 ratio)).

Fluorescence imaging was performed as previously described[18,56]. To determine the FRET efficiency of the DVL3 FlAsH sensors, coverslips with the cells were mounted using an Attofluor holder (Molecular Probes) and placed on a Zeiss inverted microscope (Axiovert200), equipped with ×63 oil objective lens and a dual-emission photometric system (Till Photonics). Cells were excited at 436 ± 10 nm using a frequency of 10 Hz with 40 ms illumination time out of a total of 100 ms. Emission of ECFP (480 ± 20 nm) and FlAsH (535 ± 15 nm), and the FRET ratio (FlAsH/ECFP) were monitored simultaneously. Fluorescence signals were detected by photodiodes and digitalized using an analog-digital converter (Digidata 1440A; Axon Instruments)[18,56]. Fluorescence intensities data were acquired using Clampex software (Axon Instruments). During measurements, cells were maintained in imaging buffer; 5 mM of 2,3-dimercapto-1-propanol (also called British anti-Lewisite—BAL) was added to the cells approximately 40 s after the recording started. Recovery of ECFP fluorescence was monitored over time and FRET efficiency was calculated as described here[18,57]. One independent experiment represents approximately 4–6 repeats (i.e. single-cell FRET signals) for each condition.

**Peptide array and other peptide reagents.** The peptide library comprised of peptides (13-mer peptides, 10 residue-overlap) with sequences derived from the IDRs of DVL3. This peptide library comprised a set of non-phosphorylated peptides and a set of phosphorylated peptides, in which the phospho groups were located at those residues corresponding to the previously identified CK1ε's phosphorylation pattern[29] (Supplementary Fig. 3). The designed peptide library was generated by JPT Peptide Technologies GmbH (Berlin, Germany) in an array format, wherein the peptides were immobilized on a glass slide.

In each experiment, two glass slides were used: one was as blank and another was screened with the recombinant CK1ε. Both slides were blocked by treatment with SmartBlock solution (Candor Bioscience, #113125) for 1 h at 30 °C. Then, one slide was incubated overnight with 10 μg/ml of the recombinant human CK1ε (MyBioSource, #MBS964562) in SmartBlock solution at 4 °C and the second one with SmartBlock solution only. The next day, both slides were washed in TBS buffer (4 × 10 min, RT) and then incubated with 1 μg/ml of the primary anti-CK1 antibody in SmartBlock solution for 1 h at 30 °C. The slides were subsequently washed again in TBS buffer (4 × 10 min, RT), followed by incubation with 1 μg/ml of secondary fluorescent antibodies in SmartBlock solution for 45 min at 30 °C, and subsequent washing in TBS buffer (4 × 10 min, RT) and in water (4 × 10 min, RT). To remove the excess of water, the slides were spinned down using a swing rotor (1200 r.c.f. for 2 min, RT, slides placed in 50 ml falcons). Binding of CK1ε to the peptides was detected by reading the fluorescence intensity of the peptide spots. In these experiments, the primary anti-CK1ε antibodies used were from Santa Cruz Biotechnology (#sc-6471) and from BD Biosciences (#610445), and the secondary antibodies used were: Alexa Fluor 594 Goat Anti-Mouse IgG2a (γ2a) (Invitrogen,

#A-21135) and Alexa Fluor 594 Donkey Anti-Goat IgG (H + L) (Invitrogen, #A-11058). The slide data were analyzed using the PepSlide Analyzer software (Sicasys).

In each experiment, the signal intensity from each peptide on the experimental set of two slides (i.e. the one used as blank and the one screened with CK1ε) was normalized to 0–100%, where the 100% value corresponded to the strongest signal detected for one of the peptide spots within the CK1ε-screened slide. The intensities measured for the peptide spots on the control slide—which was incubated with the detection antibodies only—were then subtracted from the intensities measured for the corresponding peptide spots on the CK1ε-screened slide. Each experiment was repeated 2–3 times (2–3 technical replicates per experiment) using the same anti-CK1ε antibody and average values for each antibody were added up and plotted in the graph. The signals with the highest intensity involving at least four consecutive peptides were considered as a putative CK1ε-binding site. The signals shown in the graph (Fig. 3b) correspond to the non-phosphorylated peptides and their numbering is arbitrary.

The peptides used in the fluorescent polarization (FA) binding assays were synthesized at Pepscan B.V. (Lelystad, Netherlands). The sequences of these peptides (indicated in Supplementary Fig. 4a) were amidated at their C-termini, and their N-termini were acylated with 6-aminocaproic acid (Ahx) modified with fluorescein isothiocyanate (FITC). The peptides used in the NMR/SPR studies were purchased from KareBay Biochem, Inc. (New Jersey, USA). The peptides for the NMR/SPR studies contained no terminal modifications.

**Fluorescence anisotropy.** Data were recorded at 37 °C using Tecan Infinite F500 (Tecan Systems, Germany). FITC-labeled peptides were dissolved in DMSO and added to 384-well plates (Corning) to a total volume of 20 μl (peptide final concentration was 50 nM). Incubation with recombinant His-tagged human CK1ε (MyBioSource, #MBS964562) was performed for 30 min at 37 °C. Then, fluorescence intensities were collected simultaneously by detection at 535/590 and 485/535 nm excitation/emission wavelengths, respectively, followed by adjustment to the blank controls.

**Mass spectrometry.** Samples were loaded onto SDS-PAGE gels, separated, and fixed with acetic acid in methanol, stained with Coomasie brilliant blue for 1 h and partially destained. Corresponding 1-D bands were excised. After destaining, the proteins in gel pieces were incubated with 10 mM DTT at 56 °C for 45 min. After removal of DTT excess samples were incubated with 55 mM IAA at room temperature in darkness for 30 min, then alkylation solution was removed, and gel pieces were hydrated for 45 min at 4 °C in digestion solution (5 ng/μl trypsin, sequencing grade, Promega, Fitchburg, Wisconsin, USA, in 25 mM AB). The trypsin digestion proceeded for 2 h at 37 °C on Thermomixer (750 r.p.m.; Eppendorf, Hamburg, Germany). Subsequently, the tryptic digests were subsequently cleaved by chymotrypsin (5 ng/μl, sequencing grade, Roche, Basel, Switzerland, in 25 mM AB) for 2 h at 37 °C. Digested peptides were extracted from gels using 50% ACN solution with 2.5% formic acid and concentrated in a speedVac concentrator (Eppendorf, Hamburg, Germany). The aliquot (1/10) of concentrated sample was transferred to LC-MS vial with already added poly-ethylene glycol (PEG; final concentration 0.001%)[58] and directly analyzed by LC-MS/MS for protein identification.

The rest of the sample was used for phosphopeptide analysis. Sample was diluted with acidified acetonitrile solution (80% ACN, 2% FA). Phosphopeptides were enriched using Pierce Magnetic Titanium Dioxide Phosphopeptide Enrichment Kit (Thermo Scientific, Waltham, Massachusetts, USA) according to the manufacturer's protocol and eluted into LC-MS vial with already added PEG (final concentration 0.001%). Eluates were concentrated under vacuum and then dissolved in water and 0.6 μl of 5% FA to get 12 μl of peptide solution before LC-MS/MS analysis.

LC-MS/MS analyses of peptide mixture were done using the RSLCnano system connected to an Orbitrap Elite hybrid spectrometer (Thermo Fisher Scientific) with ABIRD (Active Background Ion Reduction Device; ESI Source Solutions) and Digital PicoView 550 (New Objective) ion source (tip rinsing by 50% acetonitrile with 0.1% formic acid) installed. Prior to LC separation, tryptic digests were online concentrated and desalted using trapping column (100 μm × 30 mm) filled with 3.5-μm X-Bridge BEH 130 C18 sorbent (Waters). After washing of trapping column with 0.1% FA, the peptides were eluted (flow 300 nl/min) from the trapping column onto Acclaim Pepmap100 C18 column (3 μm particles, 75 μm × 500 mm; Thermo Fisher Scientific) by 65 min long gradient. Mobile phase A (0.1% FA in water) and mobile phase B (0.1% FA in 80% acetonitrile) were used in both cases. The gradient elution started at 1% of mobile phase B and increased from 1% to 56% during the first 50 min (30% in the 35th and 56% in 50th min), then increased linearly to 80% of mobile phase B in the next 5 min and remained at this state for the next 10 min. Equilibration of the trapping column and the column was done prior to sample injection to sample loop. The analytical column outlet was directly connected to the Digital PicoView 550 ion source.

MS data were acquired in a data-dependent strategy selecting up to top six precursors based on precursor abundance in the survey scan (350–2000 *m/z*). The resolution of the survey scan was 60,000 (400 *m/z*) with a target value of $1 \times 10^6$ ions, one microscan, and maximum injection time of 200 ms. High-resolution (resolution 15,000 at 400 *m/z*) HCD MS/MS spectra were acquired with a target

value of 50,000. Normalized collision energy was 32% for HCD spectra. The maximum injection time for MS/MS was 500 ms. Dynamic exclusion was enabled for 45 s after one MS/MS spectra acquisition and early expiration was disabled. The isolation window for MS/MS fragmentation was set to 2 $m/z$.

The analysis of the mass spectrometric RAW data files was carried out using the Proteome Discoverer software (Thermo Fisher Scientific; version 1.4) with in-house Mascot (Matrixscience; version 2.4.1) search engine utilization. MS/MS ion searches were done against in-house database containing expected protein of interest with additional sequences from cRAP database (downloaded from http://www.thegpm.org/crap/). Mass tolerance for peptides and MS/MS fragments were 7 ppm and 0.03 Da, respectively. Oxidation of methionine, deamidation (N, Q), and phosphorylation (S, T, Y) as optional modification, carbamidomethylation of C as fixed modification, and three enzyme miss cleavages were set for all searches. The phosphoRS feature was used for phosphorylation localization.

Quantitative information was assessed and manually validated in Skyline software (Skyline daily 3.1.1.8884).

**Numerical data and statistics.** As for numerical data, mean ± S.D. (or median ± interquartile range, respectively) are shown as depicted. No pairing one-way ANOVA test with Gaussian distribution (with Tukey's post-test) was used for comparison of more than two samples; for two-column statistics in Figs. 3e, g and 5b, unpaired two-tailed Student's $t$-test was used (*, $p \leq 0.05$; **, $p \leq 0.01$; ***, $p \leq 0.001$, ****, $p \leq 0.0001$; ns, not significant, $p > 0.05$).

Data on frequency of puncta vs. even (Fig. 8b) were analyzed using Generalized Linear Mixed Model (GLMM) with binomial distribution of dependent variable, treatment as fixed effect, and experiment as random effect variable, since data within the experiments are not independent. In case of evaluation of DVL-membrane recruitment results (Fig. 8c), the GLMM with multinomial distribution was used. Results were presented as odds ratios (OR) with 95% confidence intervals. Data were analyzed in R software. All *numerical* data were analyzed by MS Excel 2007 and arranged in the graphs using GraphPad Prism 6 software.

**Multiple sequence alignment.** The multiple sequence alignment of selected sequences was performed by ClustalW algorithm. Output alignment was refined manually using the BioEdit v7.0.1 sequence editor.

**Protein expression and purification.** The PDZ domain of human DVL2 wild type (aa 265–361) and phosphorylation-mimicking variant S286E + S329E were cloned into pET vector containing an N-terminal His6-tag and a lipoyl domain separated by a Tobacco Etch Virus (TEV) protease digestion site from the N-terminus of the inserts. Both proteins were expressed at high yields in *E. coli* BL21 (DE3) strain (New England Biolabs, #C2527I). For the NMR studies, 13C- and 15N-labeled proteins were prepared by growing cells in minimal medium supplemented with 15NH4Cl (1 g/l) and 13C6 glucose (2 g/l) as the sole nitrogen and carbon sources, respectively. Cells were grown at 37 °C to an OD600 of approximately 0.8 and protein expression was induced with 0.5 mM IPTG. Cells were lysed by sonication and centrifuged at 15,000g. Three purification steps were used for both proteins: Ni-NTA chromatography followed by overnight TEV cleavage to remove the N-terminal His6-tag and the lipoyl domain, ion exchange chromatography, and size exclusion chromatography. Finally, the untagged PDZ domain wild type or phosphorylation-mimicking variant was concentrated and exchanged to buffer for NMR or binding experiments containing 20 mM HEPES (pH 6.8) and 50 mM KCl. Human CK1ε (aa 1–294) was also cloned into a pET vector and purified using three purification steps as described for the PDZ domain.

Full-length DVL3 used as a substrate in the kinase assay was produced as follows: Twin-Strep-tag N-terminally tagged DVL3 was expressed in HEK293 cells using transient transfection. Cells were harvested 48 h post-transfection, resuspended in a lysis buffer (50 mM Tris, 150 mM NaCl, 10 mM KCl, 10% glycerol, pH 8) with cocktail of protease inhibitors (#11836145001, Roche) and 0.2% NP40 (#74385, Sigma). The mixture was incubated for 20 min on ice and cell lysis was enhanced by sonication. Cell lysate was cleared by centrifugation at 100,000g for 45 min at 4 °C and supernatant was loaded on Strep-Tactin Superflow high capacity column (#2-1237-001, IBA) equilibrated in the lysis buffer. The column was washed in lysis buffer and the protein was eluted using lysis buffer supplemented with 3 mM desthiobiotin. Eluted proteins were concentrated to 1 mg/ml using protein concentrators (#88516; Thermo Fisher Scientific), flash frozen in liquid nitrogen, and aliquots were stored at −80 °C.

**In vitro kinase assay.** Full-length DVL3 protein and CK1ε kinase were dialyzed into phosphorylation buffer (50 mM sodium phosphate, pH 6.5, 50 mM KCl, 10 mM MgCl2, 1 mM DTT, 1 mM EDTA). In each reaction, DVL3 and CK1ε were mixed upon 2:1 ratio (containing ~0.5 μM DVL3, ~0.25 μM CK1ε kinase, and 1 mM ATP), and incubated at 25 °C for 16 h. Samples were subsequently analyzed by SDS-PAGE and mass spectrometry.

**NMR spectroscopy.** A 4D HC(CC-TOCSY(CO))NH and a 4D 13C, 15N edited HMQC-NOESY-HSQC experiments were recorded at CEITEC Josef Dadok

National NMR Centre on a 700 MHz Bruker Avance III spectrometers equipped with 1H/13C/15N TCI cryogenic probe head with z-axis gradients. 4D spectra were processed with SSA package[59] and analyzed with SPARKY. Chemical shift assignments were obtained automatically using 4D-CHAINS[60] and checked manually. NMR titrations were performed in series of 1H, 15N HSQC spectra using 100 μM of 15N-labeled protein (wild type or phosphorylation-mimicking variant) and increasing amounts of C-terminal DVL peptide (pS700) consisted of aa 698–716 from DVL3 (stock concentration of 800 μM). Steady-state15N-1H Nuclear Overhauser Effect values were measured under a steady-state condition with a 30 s interscan relaxation delay. Reference spectra and the spectra measured under steady-state conditions were measured in an interleaved manner.

**Surface plasmon resonance.** SPR experiments were performed on a Biacore T200 instrument (GE Healthare). The wild type and phosphorylation-mimicking variant of PDZ from DVL2 were immobilized on carboxy-methylated dextran matrix (CM5 chip S Series) using amino coupling. Non-reacted groups were blocked by 1 M ethanolamine. The blank channel was modified by NHS/EDC activation and subsequently blocked by ethanolamine. The C-terminal DVL peptide (aa 698–716 from DVL3; pS700) was injected at increasing concentrations and long enough to reach steady state. SPR responses were measured and corrected for the response of the blank. Data points were collected from each binding sensogram before the end of the injection (equilibrium) and the dissociation constants were estimated by fitting the response at equilibrium versus the C-terminal peptide concentration using the following equation: $R_{eq} = C*R_{max}/(K_D + C)$, where $R_{eq}$ is the binding level in Response Units at different C-terminal peptide concentrations ($C$), $R_{max}$ the response at saturation (fitted), and $K_D$ the apparent dissociation constant (fitted).

**In silico simulation methodology.** All simulations were done in Gromacs 5.1.2 simulation package[61]. Each simulation contained a PDZ domain in a salt solution of 150 mM NaCl. The initial structure of wild-type PDZ domain (aa 245–338) from DVL3 was constructed by homology modeling using MODELLER v9.11 (refs. [62,63]). As a template we used crystal structure of human DVL2 (PDB ID: 2REY) which share 96.28% sequence identity. Missing loops and one missing residue at C-terminus were added via MODELLER[64]. Quality of seven generated models was evaluated based on DOPE and GA341 score and the best structure has been selected for further use in molecular dynamics. Mutation of serine residues S268 and S311 to either glutamic acid (*E*) or phosphoserine were done with MODELLER. The initial structure of C-terminal peptide bound to PDZ domain was taken from the Coarse-Grained simulation using PLUM parameterization[65], where a position of the C-terminal peptide was sampled around the whole PDZ domain and most probable configuration was chosen as a starting structure.

We employed all-atom force field amber99sb-ILDN[66] with phosphoserine parameters[67] in explicit TIP3P[68] water model. Robert Best's correction for disordered proteins/regions was applied by scaling van der Waals interactions between water oxygen and protein with factor 1.1 (ref. [69]). All systems were solvated in ~10,000 water molecules placed in a cubic box, where the protein is surrounded by at least 1.3 nm from the box side. Protonation state of all residues was chosen according to neutral pH = 7 with histidine H324 neutral. If not stated otherwise, all proteins were capped at C-terminus by an acetyl group and at N-terminus by *N*-methyl to remove artificial charge introduced at protein ends.

All simulations were performed for at least 300 ns with 2 fs time step. The systems were kept in NPT ensemble at 309.15 K. Temperature was controlled using a velocity-rescaling thermostat[70] with a coupling constant of 0.1 ps applied separately to solvent and protein. The pressure was hold at 1 bar via an isotropic Parrinello–Rahman barostat[71,72], with coupling time 2 ps. Cut-off distance for direct space electrostatic and van der Waals interactions was set to 1 nm. Long-range electrostatic contribution was evaluated by using the particle-mesh Ewald summation method[73,74], with a maximal distance between FFT grid points of 0.12 nm and fourth interpolation order. All covalent bonds of hydrogen atoms were constrained with LINCS algorithm[75]. All systems were simulated using periodic boundary conditions.

The mean enthalpic energy contribution for each amino acid pair was calculated from the sum of the Lennard–Jones and Coulombic contributions for the whole residue including both backbone and side-chain atoms. Averages were taken from the whole 500 ns long production dynamics.

**Reporting Summary.** Further information on experimental design is available in the Nature Research Reporting Summary linked to this article.

## Data Availability

Data supporting the findings of this manuscript are available from the corresponding author upon reasonable request. A reporting summary for this article is available as a Supplementary Information file. The source data underlying Figs. 1d, 2a, c, d, 3e, g–h, j, 4a, c, d, 5b–e, 6h, j, 7e, f, 8b–d and Supplementary Figs 1c, e, 2a, b, 4a–d, f, 7b are provided as a Source Data file. The mass spectrometry proteomics data have been deposited to the ProteomeXchange Consortium via the PRIDE[76] partner repository with the dataset identifier PXD013085. Figures were deposited in figshare depository possessing the identifier DOI 10.6084/m9.figshare.7856168

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

## Acknowledgements

We would like to thank Lucie Nesvadbová, Lenka Bryjová, Pavlína Žofka Mrhálková, and Naďa Bílá for excellent assistance, and the members of the Bryja Lab for helpful discussions. Also, we would like to thank Petra Ovesná for help with the statistics in Fig. 8. Special thanks go to Kathyann Lee for the help with English Language editing and review. The work was supported by Czech Science Foundation (project no. GA15-21789S, GA18-17658S, and GA19-26041x), Grant Agency of Masaryk University (MUNI/G/1100/2016, MUNI/G/0739/2017, MUNI/E/0533/2018). J.H. specially thanks for the support of the EMBO Short-Term Fellowship (no. ASTF 687-2016). Furthermore, this research was supported by project CEITEC 2020 (no. LQ1601) with financial contribution from the MEYS CR and National Programme for Sustainability II to R.V., M.J., K.T., L.T., K.H. and J.R. and by the; project SYMBIT (CZ.02.1.01/0.0/0.0/15_003/0000477) funded by the European Regional Development Fund and by the MEYS CR; CIISB research infrastructure project LM2015043 funded by MEYS CR is gratefully acknowledged for the financial support of the measurements at CEITEC Josef Dadok National NMR Centre and Proteomics Core Facility. J.R. thanks for support of Ministry of Health of the Czech Republic project (no. 15-34405A). M.C.A.C. and C. Hoffmann were supported by the ITN WntsApp (grant no. 608180; www.wntsapp.eu) and the Deutsche Forschungs-gemeinschaft (DFG, German Research Foundation)—TRR166: project C2. Computational resources were provided by the CESNET LM2015042 and the CERIT Scientific Cloud LM2015085 provided under the program "Projects of Large Research, Development, and Innovations Infrastructures". This work was supported from European Structural and Investment Funds, Operational Programme Research, Development and Education – "Preclinical Progression of New Organic Compounds with Targeted Biological Activity" (Preclinprogress) -CZ.02.1.01/0.0/0.0/16_025/0007381. Additional computational resources were obtained from 'IT4 Innovations National Supercomputing Center -- LM2015070' project supported by MEYS CR from the Large Infrastructures for Research, Experimental Development and Innovations. C. Holler and A.S. were supported by Deutsche Forschungsgemeinschaft (DFG, German Research Foundation), grant SCHA965/6-2.

## Author contributions

J.H., K.T., R.V., C. Hoffmann., and V.B. designed the research. J.H. performed all the research and data visualization except those mentioned below. M.C.A.C. contributed to the FlAsH-based FRET approach; M.J. and R.V. performed the in silico simulations and corresponding data visualization; J.K. and K.T. performed the NMR and SPR analyses and corresponding data visualization; C. Holler and A.S. performed the in vivo analyses in *Xenopus* and corresponding data visualization; K.H., O.B., and Z.Z. performed the MS/MS-based analysis; K.G. contributed to the IP and WB approach; T.G. and T.W.R. cloned the cell lines (CK1ε$^{-/-}$ and ECFP-DVL3 FIII cells); M.K. cloned and purified the recombinant DVL3 protein; L.T. and J.R. contributed to the peptide array approach; Z.D. and A.I.F.-L. synthesized the peptides for the FA approach; J.H., J.K., M.J., K.H., Z.Z., R.V., A.S., K.T., C. Hoffmann, and V.B. analyzed data; J.H., R.V., K.T., and V.B. wrote the original manuscript; J.H. and V.B. revised and edited the paper.

## Additional information

**Competing interests:** The authors declare no competing interests.

