## [Peer Review File · Nature Communications]

Reviewers' Comments:

Reviewer #1:

Remarks to the Author:

The paper entitled Dishevelled-3 protein conformation dynamics analysed by FRET-based biosensors: a key role of casein kinase 1 showed using different methodological approaches the role of CK1 in DVL3 conformational change. The authors uncovered new interaction domains in DVL3 for CK1 and the role of 2 phosphorylated serine residues in regulating the open and close DVL3 conformations. The paper is very well written, the experiments are clearly explained and well illustrated. It is without a doubt a very nice piece of work and bring new and interesting informations in the Wnt signaling pathway field. The use of BAL to displace Flash is a very elegant way to calculate FRET efficiency. Beside minor points that will be detailed below, I have a major concern regarding data from figure 4.

It is pretty much obvious that the same set of data on ECFP-DVL3 FAsH III (WT form) was used in figure 4a,b and d. In order for the authors to apply proper statistics on the same data set, all 3 experiments (fig4a, a and d) need to be done at the same time. If the authors did all 3 experiments at once, then it should be mentioned in the legend. Otherwise, the authors must redo this figure so each experiment having their own wt control data set. Also, for all FRET experiments, n represents how many cells were used (as described in the figure legends) and not the number of repeats (i.e independent experiment), the authors must change that; also the authors must mention how many independent experiments (independent transfection) were done.

Minor points:

supplementary Figure 5: no d figure and therefore the figures don't match with the legend.

Introduction, page 3: typo in line 69: it is via and not vie

Reviewer #2:

Remarks to the Author:

An earlier study by Lee et al. (eLife 2015) provided biophysical and functional evidence that the Wnt effector Dishevelled adopts a 'closed' (rather than 'open') conformation if its C-terminal end undergoes an intramolecular interaction with its own PDZ domain, and that the closed conformation is important for non-canonical signaling. In this current study, Harnos et al. take this forward, by using a FRET approach to probe this conformational switching from closed to open, and examine how this is regulated by casein kinase 1epsilon (CK1e), a positive regulator of both canonical and non-canonical Wnt signaling. They develop a set of elegant FRET biosensors, to provide evidence for an intramolecular interaction of Dishevelled within living cells. Furthermore, they find that inhibition of CK1e promotes a closed conformation, whereas overexpression of this kinase favors an open conformation, which is of interest, and could in principle be published as a short report (though see below, point 1).

Harnos et al. then proceed to investigate the mechanism by which CK1e promotes the closed conformation of Dishevelled. They claim to have identified three distinct peptide motifs within unstructured linker regions of Dishevelled through which CK1e binds to Dishevelled to promote the binding of its C-terminus to its PDZ domain, but this binding appears to be blocked by CK1e-dependent phosphorylations of two PDZ residues near the binding cleft, and also of the CK1e-binding motifs themselves. They thus propose that inactive CK1e acts as a scaffold of Dishevelled to promote its closed conformation, whereas active CK1e (or absence of CK1e) promotes its open conformation (Fig. 7). Unfortunately, the biochemical and biophysical evidence that led to this rather baroque hypothesis is seriously fraught (see specific comments below). The paper is therefore not publishable in its current form.

Specific problems (list not exhaustive):

- 1) The validation of the FRET constructs relies entirely on overexpression, which is neither sensitive nor physiological (i.e. the signaling activity of Dishevelled under these conditions is Wnt-independent). This needs to be done more rigorously, ideally in complementation assays of Dishevelled null-mutant cells in which these FRET constructs are expressed at endogenous levels and thus display Wnt-dependent signaling activity (e.g. in their very own system! See Paclikova et al., 2017). Also, it appears that these FRET constructs are not very active, although it is hard to assess this properly since the authors use normalization of fold-induction in all their luciferase reporter assays (e.g. Figs. 2a, 3g, 5f).
- 2) Fig. 3c: these peptide binding assays require more rigorous controls (i.e. it is not surprising that phosphorylation of these peptides will block their binding); e.g. individual alanine point mutations in these motifs should be tested (they should abolish binding).
- 3) Fig. 3e, f, j: the deltaALL mutant still responds to CK1e (Fig. 3j) and is phosphorylated by this kinase (Fig. 3f) even though it seems not to bind to it (Fig. 3e; however, there appears to be a transfer problem in the crucial last lane of this blot). This is rather inconsistent.
- 4) Fig. 4c: why is there no Wnt-dependent phosphorylation of LRP6 pS1490 in the CK1e $-/-$ cells? This phosphorylation should depend on CK1 γ (Davidson et al., 2005), and possibly also on glycogen synthase kinase 3 (Zeng et al., 2005), both of which are still present in these cells.
- 5) Fig. 6h-k: these simulations should be deleted since this approach cannot provide compelling evidence.
- 6) The amidoblack staining (used on blots throughout the study) is inadequate as a loading control.
- 7) The activity of CK1e in dissolving Dishevelled puncta was first reported by Cong et al. (Mol Cell Biol 2004), which should be cited.

Reviewer #3:

Remarks to the Author:

The manuscript by Harnos et al investigates the conformational dynamics of the key Wnt pathway effector protein Dishevelled (DVL). FRET-based biosensors for DVL conformation are generated by inserting a short Flash-binding CCPGCC sequence in one of the DVL disordered regions in addition to an N-terminal ECFP modification. These biosensors are employed to investigate conformational alterations of DVL by the CK1e kinase that was shown previously to phosphorylate DVL and modify its self-polymerization and promote DVL-mediated Wnt signaling activity.

In a set of complementary experiments, DVL conformational changes are characterized. DVL biosensors show altered FRET activity when CK1e activity or levels are manipulated, linking to "open" and "closed" conformations. These findings stress the importance of CK1e for modulating DVL conformation. Mapping experiments revealed CK1e interaction with three distinct regions in the disordered parts of DVL, besides the already known interaction with the DEP domain (Bernatik et al JBC 2011; Kishida et al JBC 2001). Deletion of these novel CK1e binding sites drives an "open" conformation of DVL and a lack of Wnt pathway activation in DVL overexpression studies. In subsequent experiments, the role of CK1e phosphorylation in mediating a "closed" DVL conformation is further characterized. A primary role is found for two phospho-serines in the PDZ domain that attenuate the previously described backfolding of the C-terminal tail of DVL (Lee et al, eLife 2015).

The work described in the manuscript is solid and delivers valuable information to the field, such as the role of specific CK1e-mediated phosphorylated residues in mediating a "closed" DVL conformation and the design and characterization of novel biosensors that probe different DVL conformations. Unfortunately, however, the obtained insights are not sufficiently evolved to allow for a robust mechanistic conceptual advance in Wnt signaling. A main issue is that the characterized DVL conformational forms do not clearly match with subcellular localization or pathway activation. E.g. open conformations of DVL link to both pathway inactivation (CK1e

deficiency) and activation (active CK1e; phosphorylated DVL) as well as to puncta formation, and CK1e binding has different consequences for conformation when compared to CK1e phosphorylation.

Issues that need to be addressed:

What do these findings on DVL conformation mean for cells in a biological context? A link to Wnt receptor activation is lacking and thus physiological relevance remains unclear. How do DVL conformations correlate with FZD-DVL receptor complex formation?

The authors identify an "open" and a "closed" form of DVL. CK1e binding induces the closed form while phosphorylation of DVL by CK1e induces the open form. Both forms do not correlate with the ability of DVL to make puncta or not. Strikingly, CK1e expression readily disperses DVL puncta, suggesting that phosphorylation events are responsible for inducing an even distribution of DVL. How do the authors reconcile these findings?

It would be predicted that the delta-ALL Dvl3 variant still forms puncta. Is this the case?

The FRET approach does not really describe the full 'conformational landscape' of DVL, and I would propose to adapt this statement.

Fig 1b: What was the rationale to insert the CCPGCC tag at the indicated positions?

Fig. 1d: For Flash III, a considerable amount of intermolecular FRET is apparent. This means that intermolecular interactions contribute to the observed intramolecular FRET signal. This should be discussed.

Fig.4: TOP flash assay should be done for CK1e^{-/-} cells.

Minor points:

Fig 2a,b and c: FLAG-tagged Dvl3 wt is shown here, while the text on p5 suggest these experiments concerns ECFP-DVL3 wt. This is confusing.

Why were disordered regions only used to test for CK1e binding? The rationale needs a better explanation in the results section

Fig. 3i, j: the text on page 7, top says: 'we have generated a Xenopus Dvl3 deltaALL mutant...'. This is not correct as these experiments were done using mRNA injections. Text and figure legend need to be adapted.

Suppl Fig 5: reference to Fig S5 on page 9 is incorrect (S5e should be 5f and 5f should be 5g)

Throughout the manuscript, articles are misplaced or lacking and there are some spelling and grammar mistakes. Some examples:

Page 9: Apart of = apart from

Page 4, bottom page: 'intramolecular' should be 'intermolecular'

p.8: 'non-phosphorylable' should be 'non-phosphorylatable'

The paragraph on page 12 'Our data do not provide definite....signalosome.' is hard to read.

Response to editor for re-submission:

We would like to take this opportunity to present the significantly expanded version of our original manuscript – **Dishevelled-3 protein conformation dynamics analyzed by FRET-based biosensors: a key role of casein kinase 1 (by Harnos *et al.*)**, which was submitted to Nature Communications in February 2018. All reviewers' comments and suggestions were carefully considered and found very valuable and helpful during the revision of our manuscript. Detailed responses to the comments of individual reviewers are provided below:

Reviewer #1 (Remarks to the Author):

The paper entitled Dishevelled-3 protein conformation dynamics analysed by FRET-based biosensors: a key role of casein kinase 1 showed using different methodological approaches the role of CK1 in DVL3 conformational change. The authors uncovered new interaction domains in DVL3 for CK1 and the role of 2 phosphorylated serine residues in regulating the open and close DVL3 conformations. The paper is very well written, the experiments are clearly explained and well illustrated. It is without a doubt a very nice piece of work and bring new and interesting information in the Wnt signaling pathway field. The use of BAL to displace Flash is a very elegant way to calculate FRET efficiency.

Response: We thank Reviewer 1 for the positive evaluation of our manuscript.

Beside minor points that will be detailed below, I have a major concern regarding data from figure 4. It is pretty much obvious that the same set of data on ECFP-DVL3 FAsH III (WT form) was used in figure 4a,b and d. In order for the authors to apply proper statistics on the same data set, all 3 experiments (fig4a, a and d) need to be done at the same time. If the authors did all 3 experiments at once, then it should be mentioned in the legend. Otherwise, the authors must redo this figure so each experiment having their own wt control data set. Also, for all FRET experiments, n represents how many cells were used (as described in the figure legends) and not the number of repeats (i.e. independent experiment), the authors must change that; Also, the authors must mention how many independent experiments (independent transfection) were done.

Response: We agree with the reviewer and have re-done this figure. In the revised Fig. 4 (and also in the other datasets presenting FRET data), we now show (i) only the controls run at the same time as the experimental treatments used in the panel, and (ii) indicate both the number of analyzed cells (that equals to the number of wells) and the number of independent transfections. We have improved the description and added the following sentence to the relevant Figure legends: *“One data point corresponds to one analyzed cell; datapoints from X independent transfections were merged.”*

Minor points:

supplementary Figure 5: no d figure and therefore the figures don't match with the legend.

Response: In the revised version the original Supplementary Fig. 5 has been redone and the current legend matches the figure.

Introduction, page 3: typo in line 69: it is via and not vie

Response: Corrected.

Reviewer #2 (Remarks to the Author):

An earlier study by Lee et al. (eLife 2015) provided biophysical and functional evidence that the Wnt effector Dishevelled adopts a 'closed' (rather than 'open') conformation if its C-terminal end undergoes an intramolecular interaction with its own PDZ domain, and that the closed conformation is important for non-canonical signaling. In this current study, Harnos et al. take this forward, by using a FRET approach to probe this conformational switching from closed to open, and examine how this is regulated by casein kinase 1epsilon (CK1e), a positive regulator of both canonical and non-canonical Wnt signaling. They develop a set of elegant FRET biosensors, to provide evidence for an intramolecular interaction of Dishevelled within living cells. Furthermore, they find that inhibition of CK1e promotes a closed conformation, whereas overexpression of this kinase favors an open conformation, which is of interest, and could in principle be published as a short report (though see below, point 1).

Harnos et al. then proceed to investigate the mechanism by which CK1e promotes the closed conformation of Dishevelled. They claim to have identified three distinct peptide motifs within unstructured linker regions of Dishevelled through which CK1e binds to Dishevelled to promote the binding of its C-terminus to its PDZ domain, but this binding appears to be blocked by CK1e-dependent phosphorylations of two PDZ residues near the binding cleft, and also of the CK1e-binding motifs themselves. They thus propose that inactive CK1e acts as a scaffold of Dishevelled to promote its closed conformation, whereas active CK1e (or absence of CK1e) promotes its open conformation (Fig. 7). Unfortunately, the biochemical and biophysical evidence that led to this rather baroque hypothesis is seriously fraught (see specific comments below). The paper is therefore not publishable in its current form.

Response: We would like to thank the reviewer for the comments. The model shown in the original article was made as an attempt to summarize all the obtained data. In the revised version we have removed the model that could mislead the readers but added significant amount of additional data. Further results provide stronger evidence both to the biochemical part as well as to the analysis of functional consequences of the regulation of “open” and “closed” conformation. For the list of additions see below.

Specific problems (list not exhaustive):

1) The validation of the FRET constructs relies entirely on overexpression, which is neither sensitive nor physiological (i.e. the signaling activity of Dishevelled under these conditions is Wnt-independent). This needs to be done more rigorously, ideally in complementation assays of Dishevelled null-mutant cells in which these FRET constructs are expressed at endogenous levels and thus display Wnt-dependent signaling activity (e.g. in their very own system! See Paclikova et al., 2017).

Response: In order to address this point, we have generated stable cell lines inducibly expressing DVL3 FRET constructs (both in wild type as well as DVL-null cells) at the level that is close to endogenous DVL3. This allowed us to demonstrate that FRET sensors are efficiently phosphorylated by endogenous CK1 in response to Wnt ligands and to analyze Dishevelled conformation changes in response to Wnts. Data from this experiment are presented as a novel Figure 7.

Also, it appears that these FRET constructs are not very active, although it is hard to assess this properly since the authors use normalization of fold-induction in all their luciferase reporter assays (e.g. Figs. 2a, 3g, 5f).

Response: The FRET constructs have been validated in three independent functional assays – i.e. (i) TopFlash reporter, where they all show both basal activity blocked by CK1 inhibitor as well as the capacity to be activated by CK1e, (ii) phosphorylation assay and (iii) subcellular localization assay. We have not observed in any of these assays the difference from the wild type FLAG-DVL3 that has been well characterized before and used widely in the literature {e.g. (Angers, Thorpe et al. 2006), (Bernatik, Ganji et al. 2011), (Bryja, Gradl et al. 2007)}. Please compare Fig. 2a – FLAG-DVL3 - with Suppl. Fig. 2a – FRET sensors. That is why we are convinced that addition of FLASH tags did not change the activity of DVL3 active.

2) Fig. 3c: these peptide binding assays require more rigorous controls (i.e. it is not surprising that phosphorylation of these peptides will block their binding); e.g. individual alanine point mutations in these motifs should be tested (they should abolish binding).

Response: Peptide binding assays were used to demonstrate that the peptides identified by the array indeed efficiently bind to CK1 ϵ . Phosphorylated peptides in this case served as a good negative control as they showed decreased binding to CK1 ϵ in the peptide array (Suppl. Fig. 4e in the current version). However, we agree with the Reviewer that the peptide binding assays represent only a supporting line of evidence for other interaction data. Thus, they have been moved into the Supplementary figures (Suppl. Fig. 4a in the current version).

3) Fig. 3e, f, j: the deltaALL mutant still responds to CK1 ϵ (Fig. 3j) and is phosphorylated by this kinase (Fig. 3f) even though it seems not to bind to it (Fig. 3e; however, there appears to be a transfer problem in the crucial last lane of this blot). This is rather inconsistent.

Response: In the revised version we have quantified the experiments with deltaALL mutants and could show that the reduced binding of CK1 ϵ (current Fig. 3e) and reduced phosphorylation (current Fig. 3g) are statistically significant. The *in vivo* experiment in Fig. 3j (in the original version), mentioned by the referee (now Fig. 3j), could have misled the reviewer, since *Xenopus* embryos do also respond to CK1 ϵ overexpressed alone (probably via endogenous Dvl). To clarify this point, we have added a new control column, not initially included, in the Figure 3j to show the activation of double axis by CK1 ϵ itself.

4) Fig. 4c: why is there no Wnt-dependent phosphorylation of LRP6 pS1490 in the CK1 ϵ -/- cells? This phosphorylation should depend on CK1 γ (Davidson et al., 2005), and possibly also on glycogen synthase kinase 3 (Zeng et al., 2005), both of which are still present in these cells.

Response: CK1 ϵ -deficient cells that are reported here for the first time, have been thoroughly validated using sequencing as well as a standard readout panel. Phosphorylation of LRP6 at S1490 was used as one of the readouts to demonstrate the deficiency of this cell line in response to Wnt ligands. The focus of this study is not the analysis of the CK1 ϵ – pS1490-LRP6 crosstalk. We, however, would like to argue with the reviewer that the reduced phosphorylation of S1490-LRP6 is entirely surprising: (i) it has been shown that treatment of cells with CK1-specific inhibitors reduced pS1490-Lrp6 after addition of Wnt-3a (Bernatik, Ganji et al. 2011) and that (ii) DVL acts upstream of LRP6 in the Wnt signal initiation cascade (Bilic, Huang et al. 2007, Zeng, Huang et al. 2008). As such, one can speculate that the deficiency in pS1490-LRP6 initiation is a consequence of CK1 ϵ -mediated action towards DVL acting upstream.

5) Fig. 6h-k: these simulations should be deleted since this approach cannot provide compelling evidence.

Response: In line with reviewer's recommendation, we have removed the simulation of the dynamics of b2-b3 loop.

6) The amidoblack staining (used on blots throughout the study) is inadequate as a loading control.

Response: The aminoblack-stained membranes have been removed as the loading controls. Original samples were reloaded and probed for tubulin that is now used as a loading control.

7) The activity of CK1e in dissolving Dishevelled puncta was first reported by Cong et al. (Mol Cell Biol 2004), which should be cited.

Response: We agree with the referee and this reference has been added to the revised text.

Reviewer #3 (Remarks to the Author):

The manuscript by Harnos et al investigates the conformational dynamics of the key Wnt pathway effector protein Dishevelled (DVL). FRET-based biosensors for DVL conformation are generated by inserting a short Flash-binding CCPGCC sequence in one of the DVL disordered regions in addition to an N-terminal ECFP modification. These biosensors are employed to investigate conformational alterations of DVL by the CK1e kinase that was shown previously to phosphorylate DVL and modify its self-polymerization and promote DVL-mediated Wnt signaling activity.

In a set of complementary experiments, DVL conformational changes are characterized. DVL biosensors show altered FRET activity when CK1e activity or levels are manipulated, linking to “open” and “closed” conformations. These findings stress the importance of CK1e for modulating DVL conformation. Mapping experiments revealed CK1e interaction with three distinct regions in the disordered parts of DVL, besides the already known interaction with the DEP domain (Bernatik et al JBC 2011; Kishida et al JBC 2001). Deletion of these novel CK1e binding sites drives an “open” conformation of DVL and a lack of Wnt pathway activation in DVL overexpression studies. In subsequent experiments, the role of CK1e phosphorylation in mediating a “closed” DVL conformation is further characterized. A primary role is found for two phospho-serines in the PDZ domain that attenuate the previously described backfolding of the C-terminal tail of DVL (Lee et al, eLife 2015).

The work described in the manuscript is solid and delivers valuable information to the field, such as the role of specific CK1e-mediated phosphorylated residues in mediating a “closed” DVL conformation and the design and characterization of novel biosensors that probe different DVL conformations. Unfortunately, however, the obtained insights are not sufficiently evolved to allow for a robust mechanistic conceptual advance in Wnt signaling. A main issue is that the characterized DVL conformational forms do not clearly match with subcellular localization or pathway activation. E.g. open conformations of DVL link to both pathway inactivation (CK1e deficiency) and activation (active CK1e; phosphorylated DVL) as well as to puncta formation, and CK1e binding has different consequences for conformation when compared to CK1e phosphorylation.

Response: We thank Reviewer 3 for the positive evaluation of our manuscript. In the revision inspired by Reviewer’s comments, we have focused on the more functional analysis that links changes in the conformation with the biological function. We believe that the data forming two novel figures (Fig. 7 and 8) and described in detail below, lead to the significant improvement of the manuscript and provide the biological significance of our findings.

Issues that need to be addressed:

What do these findings on DVL conformation mean for cells in a biological context? A link to Wnt receptor activation is lacking and thus physiological relevance remains unclear. How do DVL conformations correlate with FZD-DVL receptor complex formation?

Response: In order to address these points, we have generated stable cell lines inducibly expressing DVL FRET sensors and tested the response of DVL to the stimulation by Wnt3a and Wnt5a. This data (novel Fig. 7) demonstrates that both Wnts open DVL conformation (and both Wnts also induce phosphorylation of DVL3 by CK1e, as reported earlier). In the subsequent analysis we further show that the variants of DVL, which are locked in the more “closed” conformation, are less efficiently recruited to the membrane by FZD (Fig. 8). This opens the possibility that Wnt stimulation, via activation of CK1e, triggers “open” conformation of DVL that subsequently promotes more efficient FZD recruitment. This scenario and current experimental limitations are thoroughly discussed in the revised version as part of the Discussion.

The authors identify an “open” and a “closed” form of DVL. CK1e binding induces the closed form while phosphorylation of DVL by CK1e induces the open form. Both forms do not correlate with the ability of DVL to make puncta or not. Strikingly, CK1e expression readily disperses DVL puncta, suggesting that phosphorylation events are responsible for inducing an even distribution of DVL. How do the authors reconcile these findings?

Response: In the revised version, we have addressed the link between “open” and “closed” conformation and subcellular localization (DVL in puncta or not) in a systematic manner. The data are presented in the novel Fig. 8. Surprisingly, we have found a strong correlation between DVL conformation and DVL subcellular localization – DVL variants that were found more “closed” by FRET show significantly more punctate phenotype and vice versa. This observation is further discussed in more detail in the revised version as part of the Discussion.

It would be predicted that the delta-ALL Dvl3 variant still forms puncta. Is this the case?

Response: See the response to the previous point.

The FRET approach does not really describe the full ‘conformational landscape’ of DVL, and I would propose to adapt this statement.

Response: In the current version, we have omitted the term “conformational landscape”, and use simply “conformation”, as requested.

Fig 1b: What was the rationale to insert the CCPGCC tag at the indicated positions?

Response: The insertion sites of the CCPGCC tags were selected based on the *in silico* prediction of the disordered parts of the protein. Based on the PONDR-FIT prediction, we have selected sites that were predicted to be locally the most disordered. We have worked with the assumption that the tag insertion into these regions has the minimal effects on the 3D structure. This is now indicated in the Fig. 1b and described in the text: *“Each DVL3 sensor contained the ECFP tag in the N-terminus and one CCPGCC internal tag located in the intrinsically disordered region (IDR) of DVL3 or in the very C-terminus of DVL3. The CCPGCC tags were inserted at the most disordered regions predicted in silico using PONDR-FIT tool (Xue, Dunbrack et al. 2010) as schematized in Fig. 1b.”*

Fig. 1d: For FlAsH III, a considerable amount of intermolecular FRET is apparent. This means that intermolecular interactions contribute to the observed intramolecular FRET signal. This should be discussed.

Response: We have implemented the following paragraph into Results section to discuss this phenomenon: *“Despite the fact that the ECFP and CCPGCC tag (labeled by FlAsH molecule) showed almost complete co-localization in dots for all four sensors (Suppl. Fig. 1b), the intermolecular FRET efficiency was negligible, except of FlAsH III (Fig. 1d, right). These results demonstrate that ECFP-DVL3 FlAsH I, II and III sensors represent a useful tool for the analysis of DVL conformation in living cells that is not affected (FlAsH I, II) or only minimally affected (FlAsH III) by intermolecular FRET due to the DVL-DVL oligomerization.”*

Fig.4: TOP flash assay should be done for CK1e^{-/-} cells.

Response: In the revised version we have performed TOP Flash assays for all the conditions analyzed by FRET (incl. CK1e^{-/-} cells requested by the reviewer). TOP Flash data are shown as part of novel Fig. 8 (and Suppl. Fig. 4f) and commented in the accompanying text.

Minor points:

Fig 2a,b and c: FLAG-tagged Dvl3 wt is shown here, while the text on p5 suggest these experiments concerns ECFP-DVL3 wt. This is confusing.

Response: We agree with the referee and have corrected the current version of the manuscript.

Why were disordered regions only used to test for CK1e binding? The rationale needs a better explanation in the results section.

Response: In order to address this point we have added the following text to the corresponding part of Results in the revised manuscript:

“We hypothesized that IDRs significantly contribute to the interaction of DVL3 and CK1ε. The binding epitopes within the IDRs are typically defined by linear peptide motifs (Xue, Dunker et al. 2010) that can be rapidly exploited by a peptide microarray technique. In this technique the IDRs of protein of interest are chopped into overlapping peptides, which are subsequently immobilized via linker on a solid surface. Therefore, we designed a peptide microarray (13-meric peptides overlapping by 10 residues) based on the IDRs of DVL3”

Fig. 3i, j: the text on page 7, top says: ‘we have generated a Xenopus Dvl3 deltaALL mutant...’. This is not correct as these experiments were done using mRNA injections. Text and figure legend need to be adapted.

Response: We agree with the referee and corrected as follows: *“We injected a dose of mRNA encoding xDvl3ΔALL (lacking aa 338-350, 609-619 and 693-705 in xDvl3) into the marginal zone of the ventral blastomeres of the four-cell stage Xenopus laevis embryos and....”*. Figure legend has been also modified.

Suppl Fig 5: reference to Fig S5 on page 9 is incorrect (S5e should be 5f and 5f should be 5g).

Response: Corrected.

Throughout the manuscript, articles are misplaced or lacking and there are some spelling and grammar mistakes. Some examples:

- Page 9: Apart of = apart from
- Page 4, bottom page: ‘intramolecular’ should be ‘intermolecular’
- p.8: ‘non-phosphorylable’ should be ‘non-phosphorylatable’
- The paragraph on page 12 ‘Our data do not provide definite....signalosome.’ is hard to read.

Response: All suggestions have been implemented in the current version of the manuscript. The manuscript has been also checked for grammar/spelling by the native speaker.

Literature:

Angers, S., C. J. Thorpe, T. L. Biechele, S. J. Goldenberg, N. Zheng, M. J. MacCoss and R. T. Moon (2006). "The KLHL12-Cullin-3 ubiquitin ligase negatively regulates the Wnt-beta-catenin pathway by targeting Dishevelled for degradation." *Nat Cell Biol* **8**(4): 348-357.

Bernatik, O., R. S. Ganji, J. P. Dijksterhuis, P. Konik, I. Cervenka, T. Polonio, P. Krejci, G. Schulte and V. Bryja (2011). "Sequential activation and inactivation of Dishevelled in the Wnt/beta-catenin pathway by casein kinases." *J Biol Chem* **286**(12): 10396-10410.

Bilic, J., Y. L. Huang, G. Davidson, T. Zimmermann, C. M. Cruciat, M. Bienz and C. Niehrs (2007). "Wnt induces LRP6 signalosomes and promotes dishevelled-dependent LRP6 phosphorylation." *Science* **316**(5831): 1619-1622.

Bryja, V., D. Gradl, A. Schambony, E. Arenas and G. Schulte (2007). "Beta-arrestin is a necessary component of Wnt/beta-catenin signaling in vitro and in vivo." *Proc Natl Acad Sci U S A* **104**(16): 6690-6695.

Xue, B., R. L. Dunbrack, R. W. Williams, A. K. Dunker and V. N. Uversky (2010). "PONDR-FIT: a meta-predictor of intrinsically disordered amino acids." *Biochim Biophys Acta* **1804**(4): 996-1010.

Xue, B., A. K. Dunker and V. N. Uversky (2010). "Retro-MoRFs: identifying protein binding sites by normal and reverse alignment and intrinsic disorder prediction." *Int J Mol Sci* **11**(10): 3725-3747.

Zeng, X., H. Huang, K. Tamai, X. Zhang, Y. Harada, C. Yokota, K. Almeida, J. Wang, B. Doble, J. Woodgett, A. Wynshaw-Boris, J. C. Hsieh and X. He (2008). "Initiation of Wnt signaling: control of Wnt coreceptor Lrp6 phosphorylation/activation via frizzled, dishevelled and axin functions." *Development* **135**(2): 367-375.

Reviewers' Comments:

Reviewer #1:

None

Reviewer #3:

Remarks to the Author:

The authors have largely addressed my concerns and added a number of results that strengthen the manuscript and the main conclusions.

Some minor comments:

In the general description of Wnt signaling activation in *Xenopus* (p7), it would be helpful to mention upfront that injection of Dvl WT was used to induce double axis formation.

I would recommend to remove the 3D effects from the figures, they have no added value.

In figure 8a, b and c, the same wt control is shown repeatedly. The figure can be simplified and provide a better overview by just showing this reference once.

I would recommend to place back the summarizing figure (previous Figure 7) showing three scenarios: "closed" by CK1e binding, "open" by phosphorylation, and "open" by loss of CK1e (no phosphorylation). This will help the reader to understand the combined set of results.

The description of structural simulation and NMR analysis is rather long and contains too many details. This part could be simplified. In addition, although the authors mention that a native speaker checked the manuscript, some more grammar editing might help to make the manuscript more readable.

REVIEWERS' COMMENTS:

Please, see below our responses (in grey) to reviewers' comments:

Reviewer #3 (Remarks to the Author):

The authors have largely addressed my concerns and added a number of results that strengthen the manuscript and the main conclusions.

Some minor comments:

In the general description of Wnt signaling activation in *Xenopus* (p7), it would be helpful to mention upfront that injection of Dvl WT was used to induce double axis formation.

- We agree with the reviewer and implemented the following text into the manuscript:

*First, we injected a dose of mRNA encoding Dvl3 wild-type into the marginal zone of the ventral blastomeres of the four-cell stage *Xenopus laevis* embryos to induce double axis formation (Fig. 3j, right). Not surprisingly, the xDvl3 Δ ALL variant (lacking aa 338-350, 609-619 and 693-705 in xDvl3) showed dramatically reduced capacity to induce axis duplication both in the presence and absence of exogenous xCK1 ϵ (Fig. 3j, right).*

I would recommend to remove the 3D effects from the figures, they have no added value.

- We have removed the 3D effects from Figs 2, 4, 5b, 5e, 8a, 8e, 8f and 8g, as suggested.

In figure 8a, b and c, the same wt control is shown repeatedly. The figure can be simplified and provide a better overview by just showing this reference once.

- The control is shown repeatedly in order to make the figure easier to understand by clustering individual mutants/conditions to the functional groups. We would prefer to keep the figure as is in order to enhance its readability. This issue has been discussed by e-mail with the editor who approved our version.

I would recommend to place back the summarizing figure (previous Figure 7) showing three scenarios: “closed” by CK1e binding, “open” by phosphorylation, and “open” by loss of CK1e (no phosphorylation). This will help the reader to understand the combined set of results.

- We agree with the reviewer and implemented the new sketch summarizing our data as Fig. 9.

The description of structural simulation and NMR analysis is rather long and contains too many details. This part could be simplified.

- We agree with the reviewer and adapted the accompanied text in the manuscript accordingly.

In addition, although the authors mention that a native speaker checked the manuscript, some more grammar editing might help to make the manuscript more readable.

- We agree with the reviewer and the whole manuscript was again edited in detail by another native speaker with background in biology.